# High-Temperature, Bond, and Environmental Impact Assessment of Alkali-Activated Concrete (AAC)

Kruthi Kiran Ramagiri [1,*], Patricia Kara De Maeijer [2] and Arkamitra Kar [1]

1   Department of Civil Engineering, Hyderabad Campus, Birla Institute of Technology and Science-Pilani, Hyderabad 500078, India
2   Built Environment Assessing Sustainability (BEASt), EMIB, Faculty of Applied Engineering, University of Antwerp, Groenenborgerlaan 171, 2020 Antwerp, Belgium
*   Correspondence: p20170008@hyderabad.bits-pilani.ac.in

**Abstract:** Alkali-activated binders (AABs) offer the opportunity to upcycle a variety of residues into products that can have added value. Although AABs are reported to have a superior high-temperature performance, their thermal behavior is heavily governed by their microstructure. The present study, therefore, evaluates the effect of varying fly ash:slag ratios, activator modulus (Ms), and high temperatures on the microstructure of AAB using X-ray diffraction, Fourier transform infrared spectroscopy, and scanning electron microscopy coupled with energy-dispersive spectroscopy. Furthermore, the mechanical properties of alkali-activated concrete (AAC) are investigated through compressive, bond, flexural, and split tensile strengths. A life cycle assessment of AAC is performed using the ReCiPe 2016 methodology. The results from microstructural experiments show the formation of new crystalline phases and decomposition of reaction products on high temperature exposure, and they correlate well with the observed mechanical performance. The 28-days compressive strength with slag content is enhanced by 151.8–339.7%. AAC with a fly ash:slag ratio of 70:30 and Ms of 1.4 is proposed as optimal from the obtained results. The results reveal that the biggest impact on climate change comes from transport (45.5–48.2%) and sodium silicate (26.7–35.6%).

**Keywords:** alkali-activated binder (AAB); alkali-activated concrete (AAC); high temperature; sustainability; microstructure; life cycle assessment (LCA)

## 1. Introduction

Alkali-activated binders (AABs) offer the opportunity to upcycle a variety of residues into products that can have added value. Alkali-activated concrete (AAC) obtained through chemical reactions between alkalis and aluminosilicate-rich precursors is considered as a potential replacement for typical Portland cement (PC) concrete and currently used in real-life civil engineering applications [1]. AAB produced by the reaction between aluminosilicate rich industrial wastes and an alkaline activator exhibits superior mechanical and durability performance [2]. The primary advantage of AAB, in addition to comparable mechanical and durability performance to PC, is its sustainability [3]. Using AAB as an alternative binder is known to reduce the $CO_2$ emissions in the range of 9–80% [4,5]. The final performance of AAB as a binder is governed by the composition of raw materials and external conditions such as curing. Due to the diverse range of raw materials available, variation in microstructure can provide an insight into the final performance. However, there is minimal research available on correlating the microstructural changes to the specimen-level performance of AAB [6].

AAC can be considered as a suitable replacement for PC concrete when it can be successfully used for reinforced structural engineering applications. It is well-known that the most common environmental disaster that reinforced concrete can experience during its lifetime is fire. It is reported that AAC exhibits superior high-temperature performance compared to PC concrete [7]. However, in the case of reinforced concrete structures, the

bond between embedded rebar and concrete significantly governs their performance as a composite [8,9]. Bond strength influences the embedment length of reinforcement at the ultimate limit state, and it governs tension stiffening and cracks spacing, width of the structural member at service limit state [10]. Few studies were conducted on AAC using a pull-out test to validate its superior bond behavior compared to PC concrete [11–14]. The greater tensile strength of AAC and its characteristic compact interfacial transition zone (ITZ) contributes to this superior bond behavior [11,12]. Owing to a strong bond between rebar of smaller diameter and concrete, the failure was always due to the breaking of the bar rather than slippage [12], whereas specimens with larger size bars experienced failure due to the splitting of concrete. Both PC and AAC specimens failed in a brittle manner exhibiting a similar pattern of cracking [11]. The bond strengths were nearly proportional to AAC strength [13,14]. Until the peak bond stress, specimens with higher compressive strength exhibited increased slip-resistance, and beyond peak bond stress, it decreased with increasing compressive strength [15]. Curing conditions govern the bond behavior of class F fly ash-based AAC. The bond strength of AAC improved with heat curing duration [16]. The nano-silica-modified geopolymer concrete displayed superior bond performance in contrast to PC concrete, with both mild and deformed steel bars. The slip observed in deformed bars was less compared to mild steel bars for the same pull-out load [17]. However, these studies evaluated the bond behavior between embedded rebars and heat-cured AAC, with thermal curing at temperatures of 60–150 °C for 24 h [11,12].

One of the primary reasons for the shift towards alternative binders is environmental sustainability. It is reported that about 80% of the energy consumption and greenhouse gas (GHG) emissions from the building sector are caused during the operation phase [18,19]. However, the remaining 20% of the environmental burden is generated during raw material manufacturing, which can be efficiently reduced by using alternative binders that do not consume additional production energy [20]. Life cycle assessment (LCA) is an efficient technique to evaluate the environmental impact of a product throughout its life cycle. Only few extensive studies have reported the detailed environmental impact of using blended alkali-activated binders as opposed to PC in the construction industry to date.

Natural pozzolan (NP) and slag-based-AAC with a combination of sodium silicate and a combination of sodium silicate and sodium hydroxide as activators exhibited a 44.7% reduction in global warming potential (GWP) compared to PC concrete [21]. Thermal-cured NP and slag-based-AAB mortar exhibited lower GHG and $NO_x$ emissions, $SO_x$ emissions, $CO_2$ emissions, and Pb from production than PC mortar [22]. In comparison to PC, waste brick powder-based AAC with a combination of sodium silicate and sodium hydroxide as activators resulted in 63% and 81% lower energy consumption and GHG emissions, respectively. Alkali-activated concrete was successful in substantially alleviating the GHG emissions and GWP, but the other impact categories such as ozone layer depletion and ecotoxicity are not always lower than the PC counterpart. This is attributed to the use of manufactured or synthetic alkaline activators [23,24]. A combination of sodium silicate and sodium hydroxide is an extensively used activator. Sodium hydroxide promotes the dissolution of $Si^{4+}$ and $Al^{3+}$ ions from precursors to form an aluminosilicate-rich matrix, and sodium silicate contributes to soluble silicate species promoting the condensation process [25]. When used in combination, sodium silicate is used in higher content for the production of AAC with desirable mechanical performance [26]. Owing to the emissions and energy consumption associated with their manufacturing process, the conventional activator combination of sodium silicate and sodium hydroxide contributes significantly to the environmental impacts [27]. Replacing these conventional activators with residues such as waste glass, rice husk ash, and desulphurization dust lowered the emissions in the range of 60–99% [28,29]. The reduction in $CO_2$ emissions by AAB reported in the literature varies between 9–80% to that of PC [4,5].

The identified research gaps include:

- a comprehensive correlation between observed microstructural changes and the specimen-level performance of AAC when exposed to elevated temperatures,

- bond behavior of reinforced AAC after exposure to high temperatures,
- detailed environmental impact of AAC with fly ash and slag as precursors activated with a combination of sodium silicate and sodium hydroxide.

The primary objective of the present study was to investigate the mechanical performance of blended AAC before and after exposure to high temperatures and correlate the observed microstructural changes to its specimen-level performance. Eight AAC mixes with varying fly ash/slag ratio, and activator modulus were examined. The environmental impact of the proposed mixes was analyzed through LCA using the ReCiPe methodology. A simplified cost analysis to evaluate the economic viability of the proposed mixes was performed. The materials and experimental specifications adopted in the present study were discussed in the following sections.

## 2. Materials and Methods

### 2.1. Materials

The precursors used in the present study are class F fly ash procured from NTPC, Ramagundam, India and slag obtained from JSW limited, India, respectively. The procured fly ash and slag conform to the specifications of ASTM C618-19 [30] and ASTM C989/C989M-18a [31], respectively. The particle size distribution of the precursors is presented in Figure 1. Table 1 shows the oxide composition and other physical properties of the precursors.

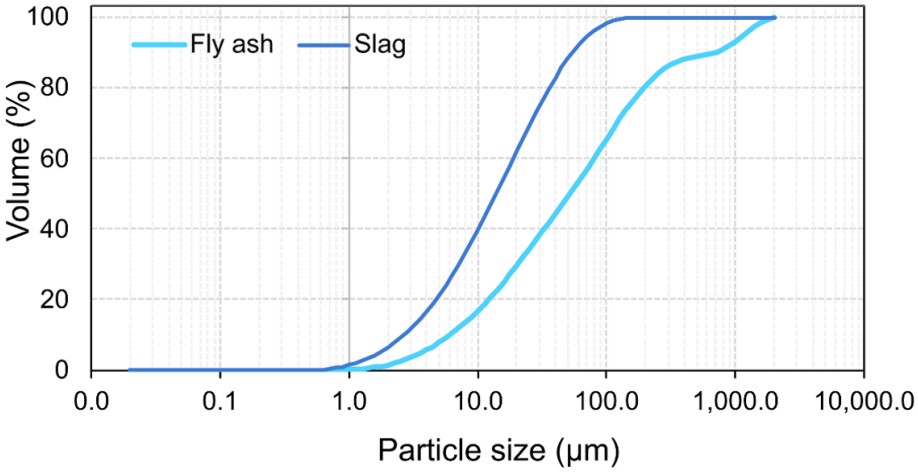

**Figure 1.** Particle size distribution of precursors.

**Table 1.** Specifications of precursors.

| Specification | Fly Ash | Slag |
|---|---|---|
| CaO (%) | 3.80 | 37.63 |
| SiO$_2$ (%) | 48.81 | 34.81 |
| Al$_2$O$_3$ (%) | 31.40 | 17.92 |
| MgO (%) | 0.70 | 7.80 |
| SO$_3$ (%) | 0.91 | 0.20 |
| Fe$_2$O$_3$ (%) | 7.85 | 0.66 |
| TiO$_2$ (%) | 2.93 | - |
| K$_2$O (%) | 1.52 | - |
| Na$_2$O (%) | 1.04 | - |
| MnO (%) | - | 0.21 |
| LOI (%) | 3.00 | 1.41 |
| Strength activity index (%) | 96.46 | 114.46 |
| d$_{50}$ (μm) | 51.90 | 13.93 |
| Specific gravity | 2.06 | 2.71 |

For the AAC mixes, the alkaline activator was prepared by mixing commercially available sodium silicate solution (33.58% $SiO_2$, 16.79% $Na_2O$, and 49.63% $H_2O$) and sodium hydroxide (food-grade, 99% purity) procured from Hychem chemicals, Hyderabad, Telangana, India. Crushed granite aggregates with a nominal maximum size of 10 mm and locally available river sand sieved through 4.75 mm, complying with the specifications of IS 383:2016 [32], were used as coarse and fine aggregates, respectively.

In the current research four distinct precursor proportions of FA: GGBFS (FS) ratios ranging from 100:0 (FS 0), 70:30 (FS 30), 60:40 (FS 40), and 50:50 (FS 50) were implemented. The denomination of FS blends was preceded by P for paste mixes and C for concrete mixes. Ms, the $SiO_2/Na_2O$ ratio of the activating solution varied as 1.0 and 1.4 in this study based on the findings of the previous study on FA-GGBFS-blended AAB [6].

The mix proportions for AAB paste and concrete are presented in Tables 2 and 3. The detailed mixing and curing procedure are elucidated in [33].

**Table 2.** Mix proportions of AAB paste.

| Mix ID | Fly Ash (kg/m$^3$) | Slag (kg/m$^3$) | Sodium Hydroxide (kg/m$^3$) | Sodium Silicate (kg/m$^3$) | Water (kg/m$^3$) |
|---|---|---|---|---|---|
| FS 0P | 1277.92 | 0 | 33.77 | 413.50 | 247.21 |
| FS 30P | 914.76 | 392.04 | 34.53 | 422.85 | 252.80 |
| FS 40P | 790.03 | 526.69 | 34.79 | 426.06 | 254.72 |
| FS 50P | 663.39 | 663.39 | 35.06 | 429.31 | 256.67 |

**Table 3.** Mix proportions of AAC *.

| Mix ID | Fly Ash (kg/m$^3$) | Slag (kg/m$^3$) | Sodium Hydroxide (kg/m$^3$) | Sodium Silicate (kg/m$^3$) | Water (kg/m$^3$) |
|---|---|---|---|---|---|
| FS 0C | 400 | 0 | 10.57 | 129.43 | 77.38 |
| FS 30C | 280 | 120 | 10.57 | 129.43 | 77.38 |
| FS 40C | 240 | 160 | 10.57 | 129.43 | 77.38 |
| FS 50C | 200 | 200 | 10.57 | 129.43 | 77.38 |

* The fine aggregate content and coarse aggregate content in AAC were kept constant at 651 kg/m$^3$ and 1209 kg/m$^3$, respectively.

### 2.2. Methods

#### 2.2.1. Exposure to Elevated Temperature

The samples were exposed to selected elevated temperatures of 538 °C, 760 °C, and 892 °C in a muffle furnace supplied by Aimil Co., Ltd. (New Delhi, India) for 2 h on the day of testing and gradually allowed to cool down to the room temperature. These temperatures were selected based on the standard time–temperature curve specified by ASTM E119-19 [34].

#### 2.2.2. Sample Preparation for Microstructural Analyses

For XRD and FTIR analyses, the samples were allowed to attain ambient temperature by natural cooling and were then finely ground using a mortar and pestle arrangement. This ground powder was sieved through a 75 μm sieve to ensure uniform particle size.

To perform SEM-EDS analysis, the sample was cooled to ambient temperature after exposure to designated elevated temperatures. This was followed by oven-drying the sample for 2 h to ensure the dehydration of free moisture. The samples were then sputter-coated with a 10 nm layer of gold-palladium using LEICA EM ACE200 sputter coater. The samples were not polished for SEM-EDS analysis owing to their tendency to disintegrate when exposed to such high temperatures and pressures. Similar sample preparation was used in previous research on AAB [35].

### 2.2.3. X-ray Diffraction

X-ray diffraction analysis was performed using RIGAKU Ultima X-ray Diffractometer. The Cu Kα X-rays were operated at 40 kV and 30 mA. The range of 2θ was maintained as 5 to 90°, with a step width of 0.02° and a scan speed of 1°/min. The recorded XRD patterns were identified through HighScore Plus software, version 4.9 by Malvern Panalytical [36].

### 2.2.4. Fourier Transform Infrared Spectroscopy (FTIR)

FTIR analyses were performed in the wavenumber range of 4000 to 400 $cm^{-1}$ using JASCO FT/IR-4200. A resolution of 4 $cm^{-1}$ with 64 scans per spectrum was employed. KBr pellet technique was used for sample preparation. The spectral bands corresponding to different molecular vibrations were recorded in transmission mode.

### 2.2.5. Scanning Electron Microscopy-Energy Dispersive X-ray Spectroscopy (SEM-EDS)

Variations in the morphology of AAB were examined using an FEI Apreo SEM setup. The corresponding changes in the elemental composition were investigated using EDS through INCA software from Oxford Instruments. Samples were oven-dried at 105 °C for 2 h prior to the analysis to remove any moisture and sputter-coated with a 10 nm layer of gold-palladium to make them electrically conductive. The images were captured at a magnification of 2500× and an operating voltage of 20 kV. The working distance and spot size were maintained as 10 mm throughout the analysis.

### 2.2.6. Compressive Strength

The test procedure conformed to the specifications of ASTM C39/C39M-18 [37] using a HEICO compression testing machine of 2000 kN capacity. The specimens (100 × 100 × 100 mm) were tested in a load control set-up at a loading rate of 0.25 ± 0.05 MPa/s.

### 2.2.7. Pull-Out Test

The bond strength of AAC was evaluated by conducting the pull-out test on the specimens with embedded rebar at the center through HEICO automated universal testing machine (UTM) of 1000-ton capacity and a specially fabricated test frame (Figure 2).

The specimens used for testing bond behavior were cast in a fabricated cube mold (100 × 100 × 100 mm) with a central hole to accommodate a vertically placed rebar. Curing conditions similar to the specimens tested for compressive strength were followed. The test procedure complied with the guidelines of the Indian standard code of practice for performing the pull-out test, IS 2770-1:2017R [38]. The tests were performed in the displacement control mode, with the loading rate maintained at 0.01 mm/s. An automated data acquisition system was used to record the load. The specimens were subjected to loading until they fail, either due to the yielding of the rebar or due to the splitting of the concrete. The bond strength reported was the average of three specimens.

The bond strength was calculated as the ratio of applied load required for the slip to the surface area of the embedded length of the rebar. The average bond stress was calculated as:

$$\sigma_{bond} = \frac{P}{\pi d L} \tag{1}$$

where,

    $d$—diameter of the reinforcement (mm)
    $L$—length of specimen parallel to the reinforcement (mm)
    $P$—load at specimen failure (N)
    $\sigma_{bond}$—average bond stress (MPa)

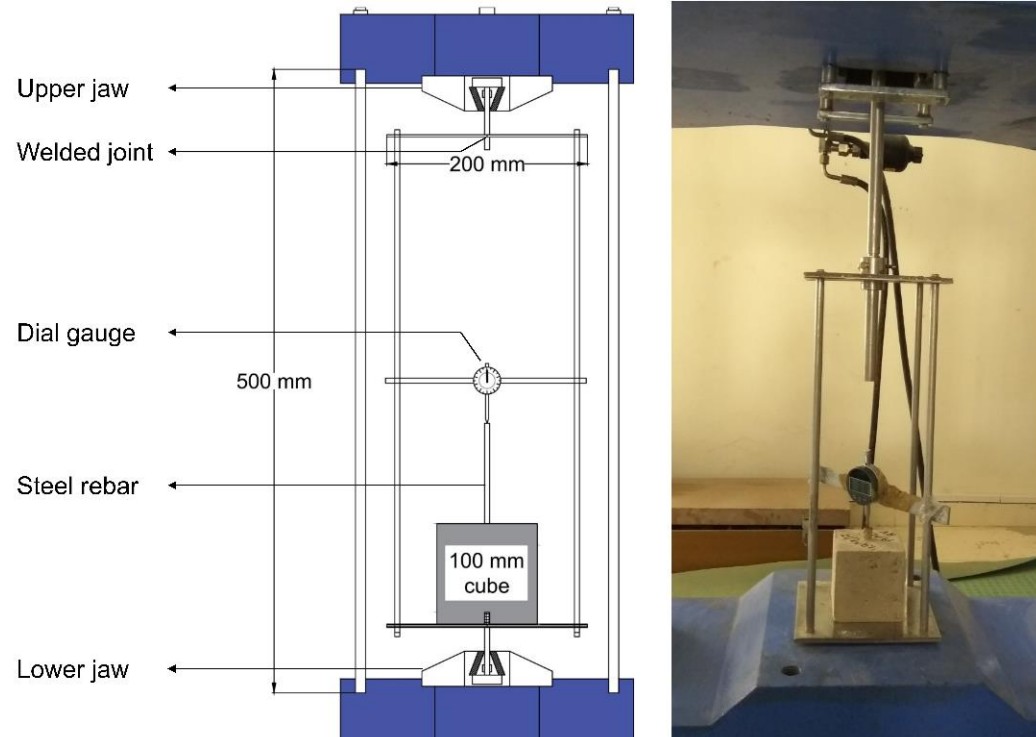

**Figure 2.** Schematic representation of pull-out test set-up.

### 2.2.8. Flexural Strength Test

The flexural strength test was performed on $100 \times 100 \times 500$ mm beam specimens to estimate the modulus of rupture. The test was performed to conform with the standard specifications of ASTM C78/C78M-18 [39]. An automated flexural testing machine of 100 kN capacity supplied by HEICO was used to perform the test.

### 2.2.9. Split Tensile Strength Test

The indirect estimation of the tensile strength of concrete was made through split tensile strength. The test was performed in a HEICO CTM of 2000 kN capacity on cylindrical specimens with a diameter of 150 mm and height of 300 mm. The test was performed to comply with the standard specifications of ASTM C496/C496M-17 [40].

### 2.2.10. Life Cycle Assessment Methodology

Life cycle assessment (LCA) is a technique to evaluate the "environmental impact of a product system throughout its life cycle", starting with the raw materials to its final disposal [41,42]. LCA was performed through four different steps: the goal and scope definition, life cycle inventory (LCI) analysis, life cycle impact assessment (LCIA), and interpretation.

### Goal and Scope

The primary aim of this study was to investigate and compare the environmental impact of AAC with varying precursor proportions and Ms. The results were analyzed to evaluate the proposed optimum mix in improving the environmental sustainability of AAC. Cost analysis was performed to assess the economic viability of the mixes in the Indian scenario. Environmental impacts were assessed for a functional unit of 1 m³ of concrete. Nine concrete mixes were assessed for their environmental impact in this study: (i) FS 0C-1.0; (ii) FS 30C-1.0; (iii) FS 40C-1.0; (iv) FS 50C-1.0; (v) FS 0C-1.4; (vi) FS 30C-1.0; (vii) FS 40C-1.4; (viii) FS 50C-1.0; and (ix) PC concrete of strength equivalent to FS 30C.

The system boundary, as presented in Figure 3, was used in the present study. The cradle-to-gate approach of LCA was adopted, including calculations of all the emissions and energy consumption from the raw material procurement to the preparation of concrete. Since the primary objective of this study was to compare different AAC mixes, the usage and disposal phases were not considered, and it was assumed that the environmental impact from these phases would be equivalent.

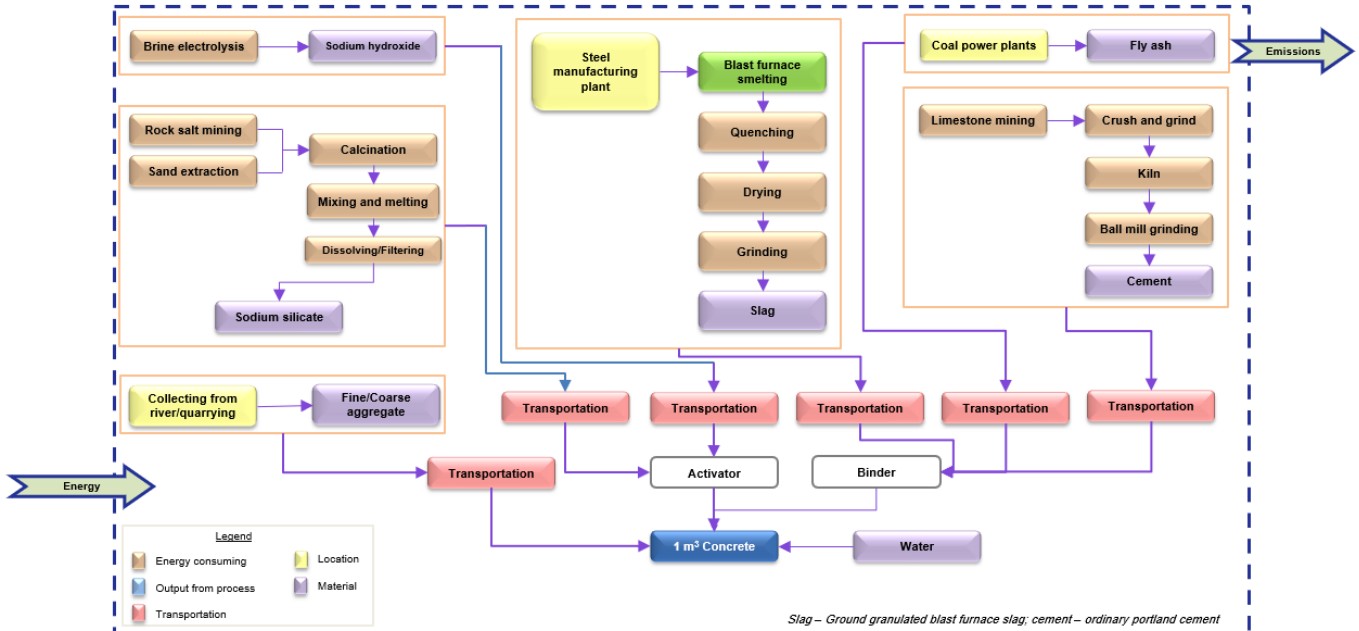

**Figure 3.** System boundary for AAC production.

Inventory Analysis

Life cycle inventory (LCI) data for the processes were obtained from Ecoinvent 3.7.1 database [43,44] and are presented in Table 4. The transportation distances of the raw materials used in the present study were calculated from their manufacturing source to Birla Institute of Technology-Pilani, Hyderabad Campus, rather than supply since the system boundary considered is from the point of production.

**Table 4.** Inventory data for raw materials.

| Raw Material | Burden Considered | Ecoinvent 3.7.1 (Rest-of-the-World) |
|---|---|---|
| Fly ash | None | - |
| Slag | Manufacturing | Ground granulated blast furnace slag |
| Cement | Manufacturing | Cement, Portland |
| Sodium silicate | Manufacturing | Sodium silicate production, hydrothermal liquor, product in 48% solution state |
| Sodium hydroxide | Manufacturing | chlor-alkali electrolysis, diaphragm cell |
| Fine aggregate | Collecting from river | Silica sand production |
| Coarse aggregate | Quarrying of stone | Gravel production, crushed |
| Water | Treatment of water | Tap water; conventional treatment |

Life Cycle Impact Assessment

Life cycle impact assessment (LCIA) was performed using the ReCiPe 2016 midpoint and endpoint methodology [45]. The hierarchist perspective was adopted in this study because it is based on a scientific agreement concerning both the time period (100 years) and the credibility of impact factors [45]. However, to verify the robustness of ReCiPe, other methods based on the ILCD handbook [46] recommendations were also used. Hence,

this study compares different impact categories evaluated using default LCIA methods and the selected ReCiPe midpoint (H) v1.13. The impact categories analyzed in this study through the ReCiPe method were GWP, fossil depletion potential (FDP), freshwater ecotoxicity (FETP), freshwater eutrophication (FEP), human toxicity potential (HTP), marine ecotoxicity potential (METP), marine eutrophication potential (MEP), ozone depletion potential (ODP), photochemical oxidant formation potential (POFP), terrestrial acidification potential (TAP), and terrestrial ecotoxicity potential (TEP). The other methods selected based on the recommendations of ILCD handbook for:

(i)     GWP is IPCC 2013 [47];
(ii)    FETP, and HTP are USEtox v1.0 [48];
(iii)   FEP and MEP are EDIP 2003 [49];
(iv)    ODP and TAP are CML 2001 [50].

The openLCA v 1.10.3, an open-source software developed by Hildenbrand et al. [51], was used to evaluate the impacts of each product within the system boundary. It has been developed by GreenDelta in Berlin, Germany.

## 3. Results and Discussion

### 3.1. X-ray Diffraction

The X-ray diffraction analysis was performed to evaluate the effect of varying precursor proportions on different phases of reaction products in AAB and analyze the variations in microstructure on exposure to high temperatures. Figure 4a,b presents the XRD pattern of AAB with Ms = 1.0 and 1.4 exposed to ambient temperature.

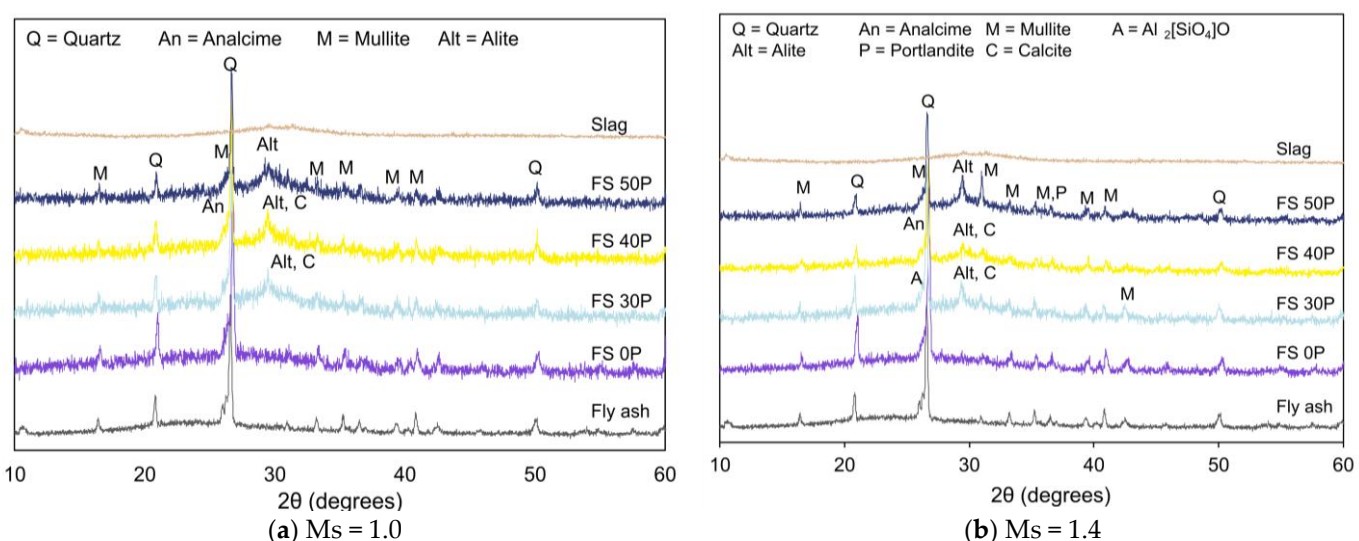

(**a**) Ms = 1.0                                            (**b**) Ms = 1.4

**Figure 4.** XRD pattern of AAB paste under ambient conditions.

The major crystalline phases of quartz ($SiO_2$) and mullite ($2Al_2O_3 \cdot SiO_2$) in the unreacted fly ash were observed in AAB. Furthermore, additional crystalline phases corresponding to analcime ($NaAlSi_2O_6 \cdot H_2O$) were formed due to the alkali-activation of the aluminosilicates. This finding was consistent with previous studies [52,53]. Analcime was observed in all AAB mixes, both Ms of 1.0 and 1.4 at 2θ values of 27°. With the addition of slag to the AAB system, peaks corresponding to alite ($3CaO \cdot SiO_2$) were observed at 2θ values of 29°, and their presence is attributed to unreacted slag. Peaks corresponding to sillimanite ($Al_2SiO_5$) and portlandite were observed in FS 30P-1.4. Peaks associated with sillimanite were observed at 2θ = 26.2°, 35.2°, and 40.8°, whereas portlandite peaks were observed at 2θ = 29.4° and 35.3°. The presence of these crystalline phases is related to the slow reactivity of fly ash. Calcite was observed in FS 30 and FS 40 mixes at 2θ values of 29.4°. The formation of calcite is attributed to the carbonation of slag-rich mixes.

Previous research on AAB mortars reveals that $CO_2$ directly reacts with calcium-rich matrix (C-A-S-H) [54].

Figure 5a,b present XRD patterns of FS 0P-1.0 and FS 0P-1.4, respectively, before and after exposure to elevated temperatures. No noticeable difference was observed in the XRD pattern of AAB (FS 0P-1.0 and FS 0P-1.4) at the ambient temperature and after exposure to 538 °C. A considerable number of new peaks were formed on exposure to higher temperatures of 760 °C and 892 °C. Geopolymeric phases tend to dehydrate at such high temperatures and recrystallize to form nepheline ($Na_3KAl_4Si_4O_{16}$), albite or other stable aluminosilicates [55]. These transformations include intermediate sintering and subsequent development of the vitreous phase [56]. With the increase in temperature beyond 760 °C, the formation of nepheline at $2\theta$ = 20.5°, 23.1°, 27.2°, 28.1°, 29.8°, 31.1°, 38.5°, and 41° can be observed from Figure 4b. At higher temperatures (892 °C), the presence of albite was also detected in AAB with Ms = 1.0. Albite (Al/Si = 0.3) is a less Al-rich phase than nepheline (Al/Si = 1) [57]. A previous study reported that both nepheline and albite were formed when $Na^+$ content is sufficient [58]. The findings from the present study corroborate with the reported research since AAB with Ms = 1.0 has higher Na+ content than AAB with Ms = 1.4.

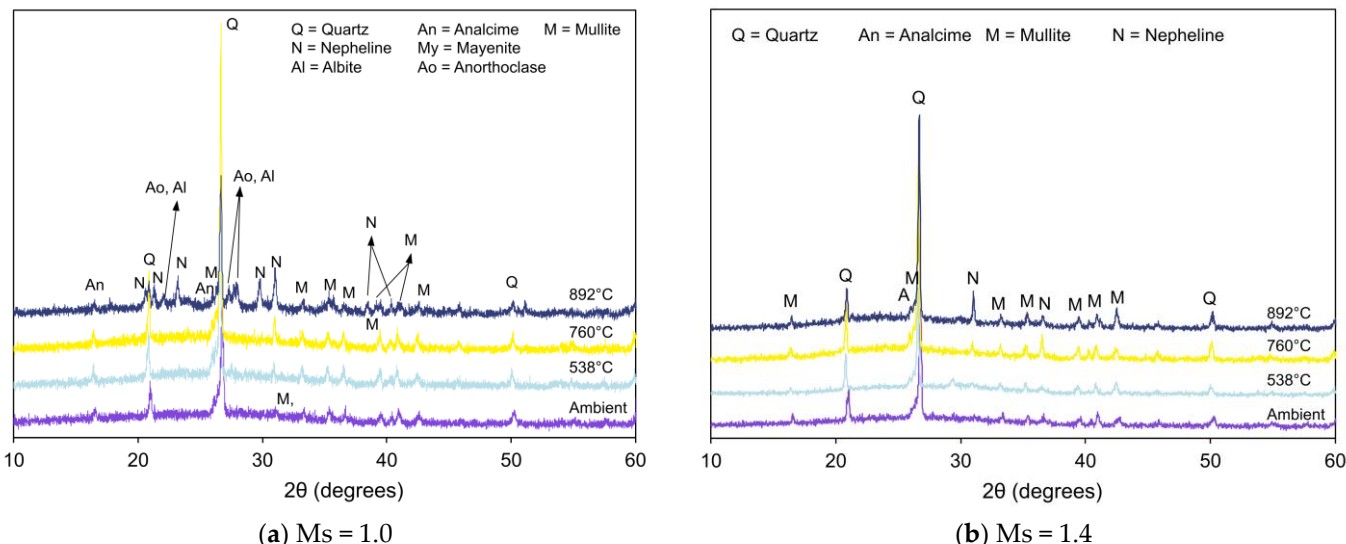

**(a)** Ms = 1.0    **(b)** Ms = 1.4

**Figure 5.** XRD pattern of FS 0P on exposure to elevated temperature.

Figures 6–8 present XRD patterns of AAB with varying fly ash:slag ratio and exposed to high temperatures [59]. With the increase in temperature to 538 °C, the absence of amorphous peaks can be observed from the diffraction patterns of FS 30P, FS 40P, and FS 50P and is congruent with its crystallization or dehydration [60]. With an increase in the exposure temperature beyond 760 °C, the X-ray diffractogram exhibits predominant peaks corresponding to new crystalline phases of akermanite, gehlenite, and nepheline. Additionally, wollastonite ($CaSiO_3$) formation was identified from the XRD patterns of FS 40P-1.0 and FS 50P-1.0. The presence of slag resulted in the growth of akermanite and gehlenite. The peaks at $2\theta$ values of 24°, 29°, 36.5°, 37.4°, and 44.4° correspond to akermanite and gehlenite; the peaks at 26.9°, 28.9°, and 30° were associated with wollastonite, and those at 17.7 °, 23.1°, 27.2°, 29.7°, 31°, 34.8°, 37.5°, and 43.2° are ascribed to nepheline.

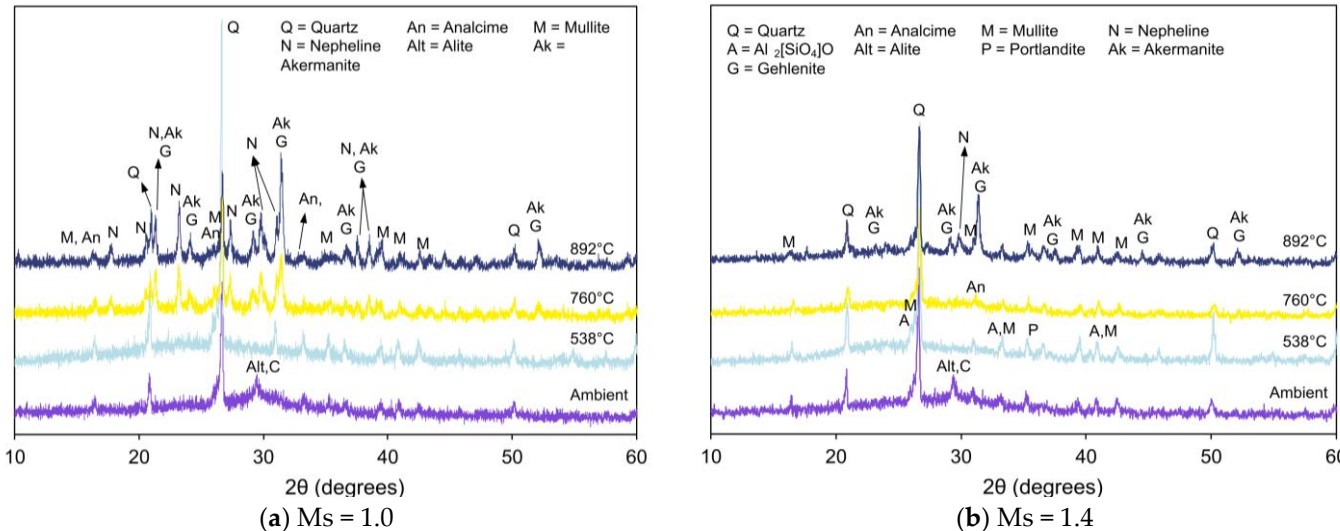

**Figure 6.** XRD pattern of FS 30P on exposure to elevated temperature.

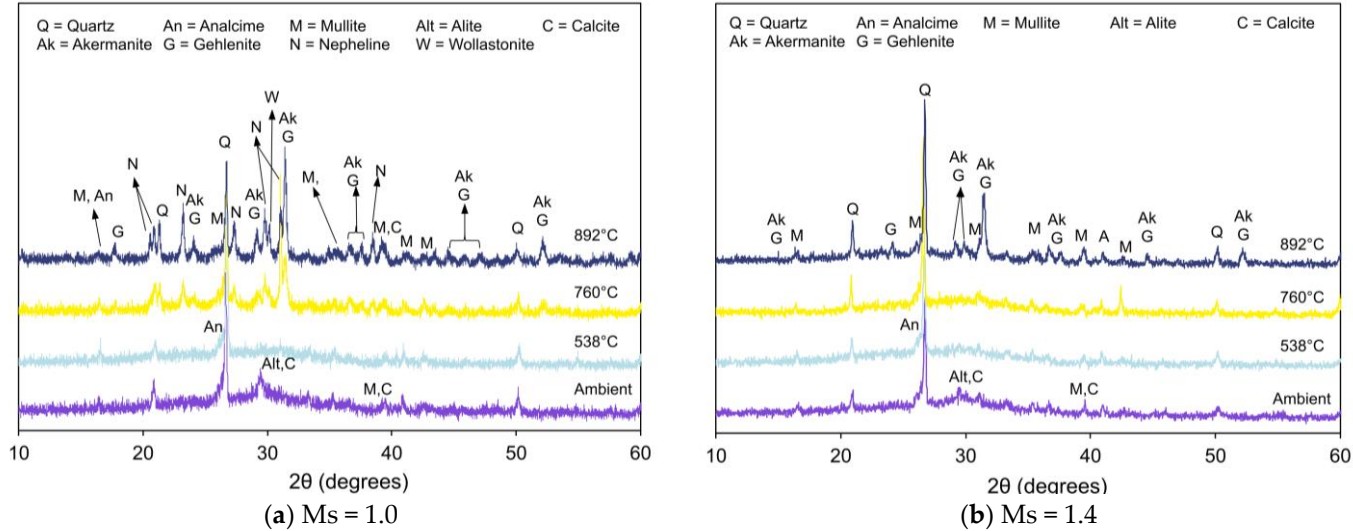

**Figure 7.** XRD pattern of FS 40P on exposure to elevated temperature.

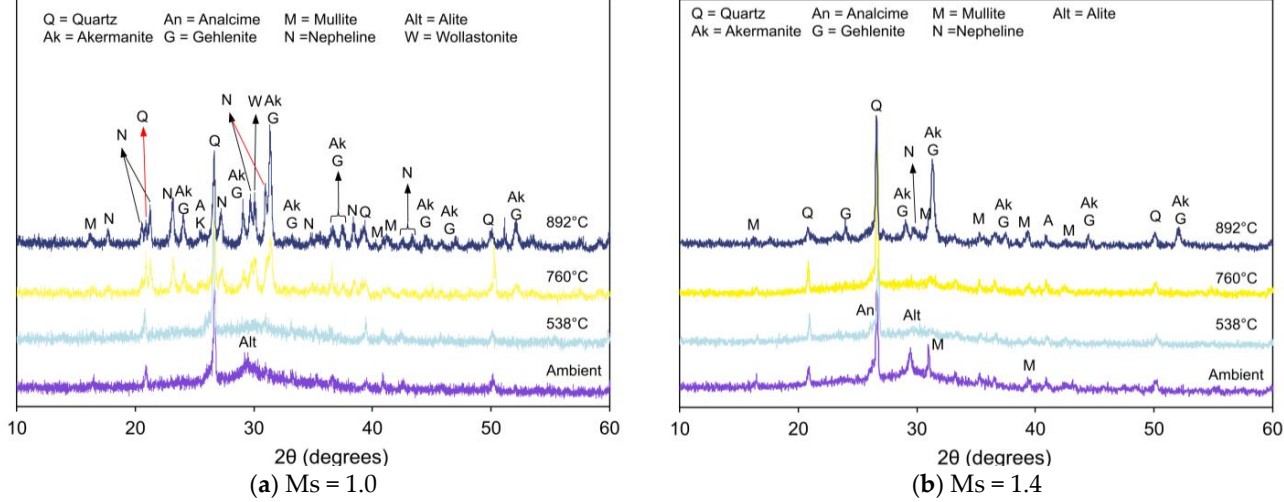

**Figure 8.** XRD pattern of FS 50P on exposure to elevated temperature.

The presence of akermanite and gehlenite cannot be differentiated from XRD analysis, as both are isomorphs with close diffraction plane spacing [61]. It should be emphasized here that the peaks corresponding to akermanite and gehlenite increase with slag content. Peaks ascribed to calcite were observed in FS 30P and FS 40P under ambient exposure conditions and were absent in the diffractograms of these samples after exposure to temperatures of 760 °C and 892 °C. This behavior is attributed to the decomposition of calcite at temperatures of 550 °C. In the present study, reflections of wollastonite were observed in XRD patterns of FS 40P and FS 50P exposed to temperatures above 760 °C. The formation of this phase is attributed to the decomposition of C-A-S-H or devitrification of slag [62].

### 3.2. Fourier Transform Infrared Spectroscopy (FTIR)

Figure 9a,b show the FTIR spectra for matured AAB samples with Ms 1.0 and 1.4, respectively, after exposure to ambient conditions. The evident disparity in the spectra of FS 0P and the other blended AAC mixes indicated the different reaction products formed on the alkali-activation of only fly ash and fly ash-slag blends. The primary reaction products formed in AAB are essentially made up of $AlO_4$ and $SiO_4$ tetrahedra. At ambient temperature, the major bands in the range of 3600–3200 $cm^{-1}$ correspond to symmetric stretching of the O-H group of water in the system. The bands in the range of 1639–1647 $cm^{-1}$ were assigned to symmetrical bending of the H-O-H bond of water. Stretching of the O-C-O band at 1411 $cm^{-1}$ ascribed to the occurrence of carbonation reaction. Enhanced carbonation resistance of AAB with a greater proportion of slag ascertains its pore refinement [63].

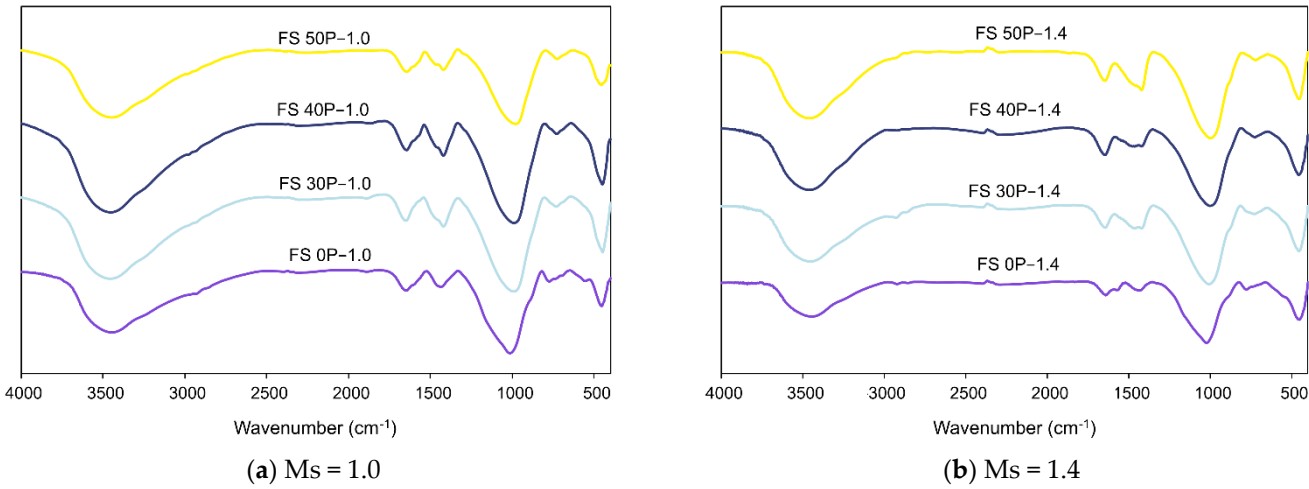

**Figure 9.** FTIR spectra of AAB paste under ambient conditions.

The bands in the vicinity of 1000 $cm^{-1}$ are assigned to the stretching of Si-O associated with SiO4 tetrahedra present in the glassy phase [64]. The chemical shift in the position of these bands with respect to FS 0P is attributed to the greater amount of Ca and the formation of C-A-S-H [52]. A previous study by Garcia-Lodeiro et al. (2011) [65] reports that the bands at 952 $cm^{-1}$ and 960 $cm^{-1}$ are characteristic of C-A-S-H and C-S-H matrices, respectively. In the present study, it is evident that with increasing slag content, the bands ascribed to N-A-S-H shifted to lower wavelengths closer to the characteristic bands of C-A-S-H and C-S-H, evincing that the formed reaction product is a combination of N-A-S-H and C-A-S-H. The band located at 777 $cm^{-1}$ is assigned to Si-O stretching of quartz and indicates the presence of unreacted fly ash. The band in the range of 460 $cm^{-1}$ corresponds to the in-plane bending of Si-O-Si linkage associated with quartz. Similar bands were observed in unreacted fly ash, and these results substantiate the XRD analysis, which showed the presence of quartz and mullite observed in fly ash before and after alkaline activation.

Figure 10a,b show the variation in the FTIR spectra of FS 0P-1.4 and FS 0P-1.0, respectively, as a function of exposed temperature.

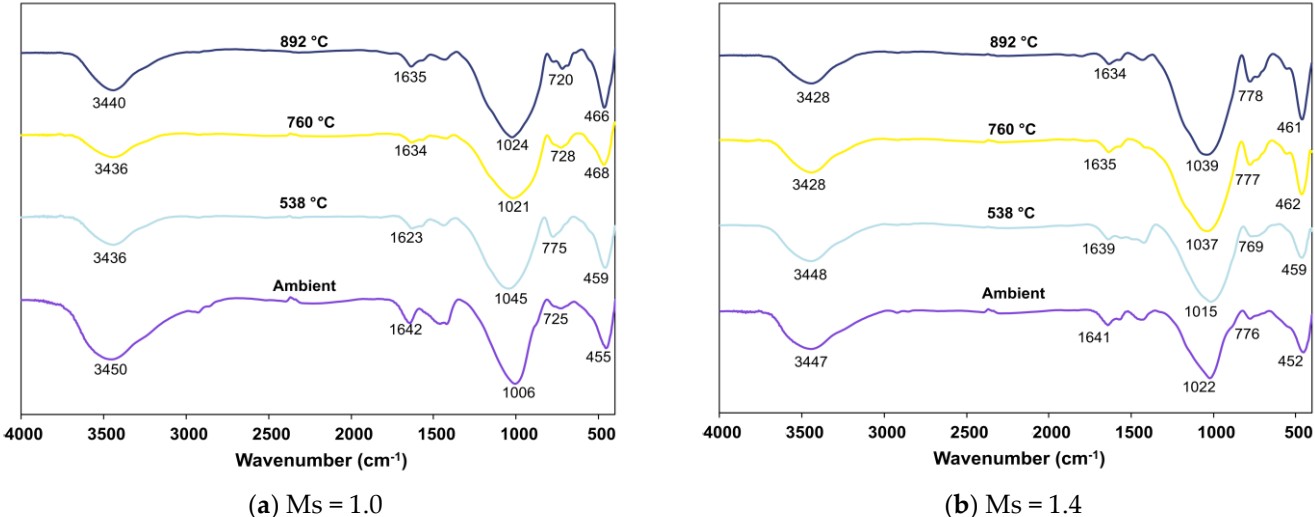

**Figure 10.** FTIR spectra of FS 0P on exposure to elevated temperature.

With the increase in temperature of exposure, the samples exhibit a gradual change in the pattern of their spectra. The perceptible changes were observed at 3600–3200 cm$^{-1}$ and 1640 cm$^{-1}$ associated with water loss from the system with increasing temperature. The other notable variation with the increase in temperature to 892 °C is the shift of bands recorded at 1013 cm$^{-1}$ to 1020 cm$^{-1}$ in FS 0P-1.0 and 1022 cm$^{-1}$ to 1039 cm$^{-1}$ in FS 0P-1.4. This shift was consistent with the formation of albite and nepheline. Upon exposure to elevated temperature, the geopolymer matrix undergoes dehydration and recrystallizes to form nepheline and albite [66]. Previous IR spectroscopic studies on pure Na-nepheline and natural albite reveal high-intensity bands in the spectral range of 996–1039 cm$^{-1}$ and 1034–1048 cm$^{-1}$ [67,68]. It is evident that with the increase in temperature to 892 °C, the bands near 452–470 cm$^{-1}$ in FS 0P shift to 461–465 cm$^{-1}$ ascertaining the formation of albite [69].

Figures 11–13 present FTIR spectra of AAB with varying fly ash:slag ratio after exposure to severely high temperatures.

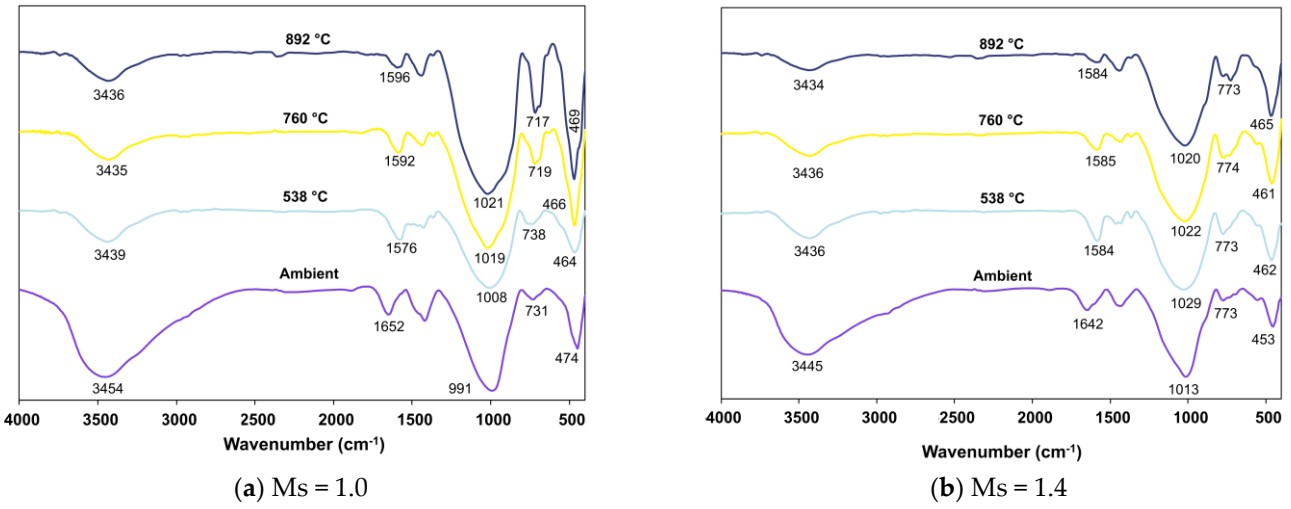

**Figure 11.** FTIR spectra of FS 30P on exposure to elevated temperature.

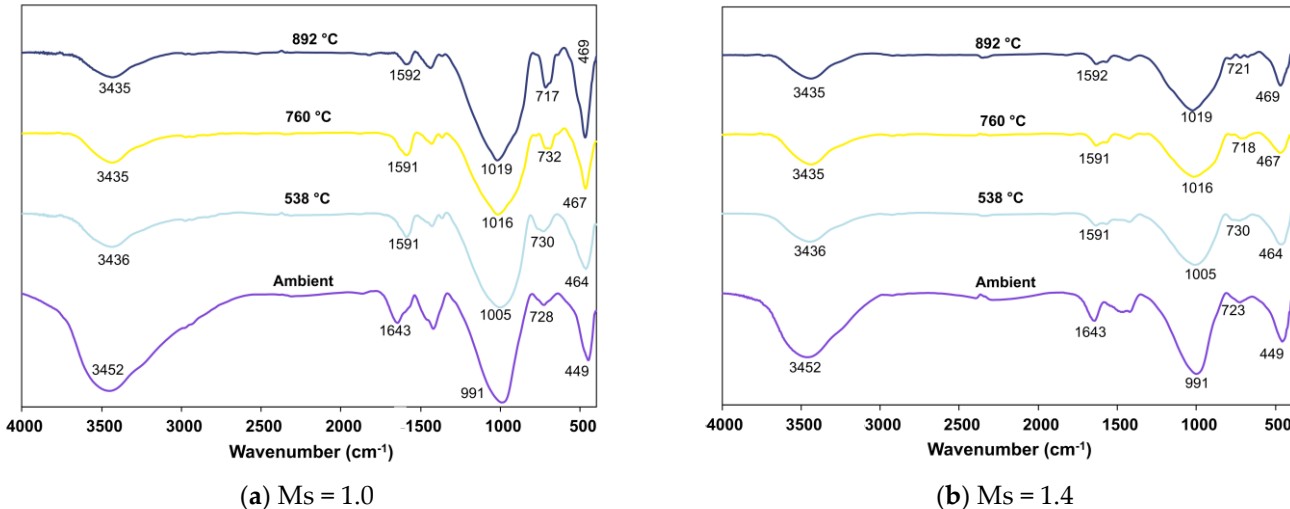

**Figure 12.** FTIR spectra of FS 40P on exposure to elevated temperature.

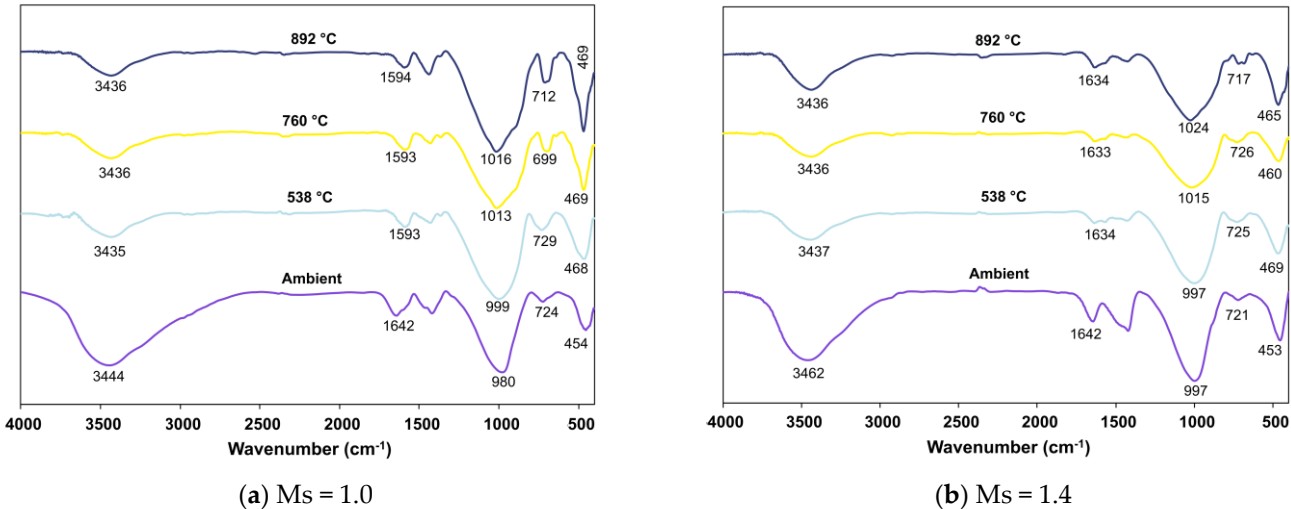

**Figure 13.** FTIR spectra of FS 50P on exposure to elevated temperature.

Analogous to the trend observed in FS 0P, the peaks corresponding to the O-H bonds of water show decreased intensity with increasing temperature in FS 30P, FS 40P, and FS 50P. An increase in exposed temperature alleviated the intensity of peaks assigned to stretching vibrations of the O-C-O bond of $CO_{3-2}$ in FS 30P, FS 40P, and FS 50P. This reduction is attributed to the carbonation of the samples on exposure to such high temperatures. The shift of bands in the range of 991 to 1024 cm$^{-1}$ was analogous to FS 0P and is attributed to the formation of nepheline. These findings corroborate with XRD results, showing the formation of nepheline in all the mixes exposed to temperatures above 760 °C. A conspicuous observation inconsistent with FS 0P mixes was the increase in the intensity of bands in the wavenumber range of 712 to 731 cm$^{-1}$ ascribed to Si-O-Si stretching.

Furthermore, in both mixes with Ms 1.0 and 1.4, these bands shift towards 712–720 cm$^{-1}$, which is assigned to Si-O-Si stretching of $SiO_4$ tetrahedra in gehlenite [70]. The intensity of FTIR spectra was proportional to the number of times a functional group occurs within the molecule, otherwise, the concentration of the molecule [71]. It is emphasized here that this increase in the intensity of bands is more in AAB systems with Ms = 1.0, indicating the formation of more gehlenite in Ms = 1.0 mixes.

### 3.3. Scanning Electron Microscopy and Energy-Dispersive X-ray Spectroscopy

The micrographs of matured AAB paste containing varying proportions of fly ash and slag and after exposure to ambient temperature are presented in Figure 14. A relatively high proportion of unreacted fly ash particles observed in the micrograph of FS 0P-1.4 is attributed to its high activation energy and absence of heat-curing (Figure 15). With the increasing proportion of slag in the mixes, reduction in the observed fly ash spheres, and precipitation of different reaction products is evident from the micrographs of FS 30P-1.4, FS 40P-1.4, and FS 50P-1.4 (Figures 16–18). In FS 0P-1.4, only spherical particles are embedded in the matrix, whereas other mixes show angular particle deposition.

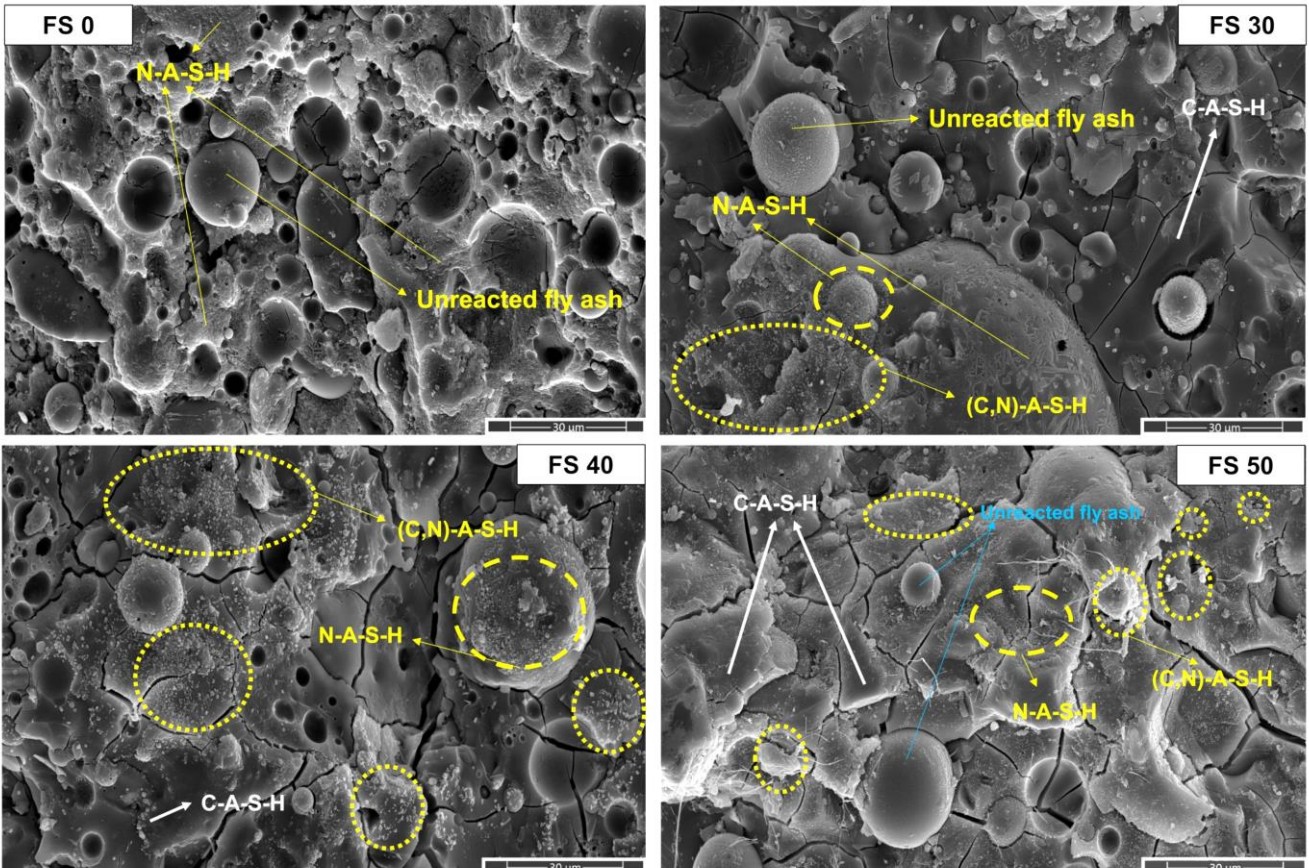

**Figure 14.** SEM micrographs of AAB with Ms 1.4 exposed to ambient temperature.

Figures 15–18 presents the SEM micrographs of AAB paste with a Ms of 1.4 and varying fly ash/slag ratio after exposure to high temperatures [72].

The morphology of AAB did not evince any perceptible changes on exposure to 538 °C, corroborating with the XRD and FTIR results. With the increase in temperature to 538 °C, the proportion of unreacted fly ash particles in FS 0P-1.4 reduced. Substantial changes in the morphology can be noticed from the micrographs of AAB exposed to 760 °C and 892 °C. With a rise in temperature to 760 °C and 892 °C, AAB mixes with higher fly ash content exhibit increased porosity. These pores provided an escape route for the moisture at high temperatures alleviating the deterioration due to pore pressure [73]. The black regions in the micrographs are associated with the pores formed due to the evaporation of free water.

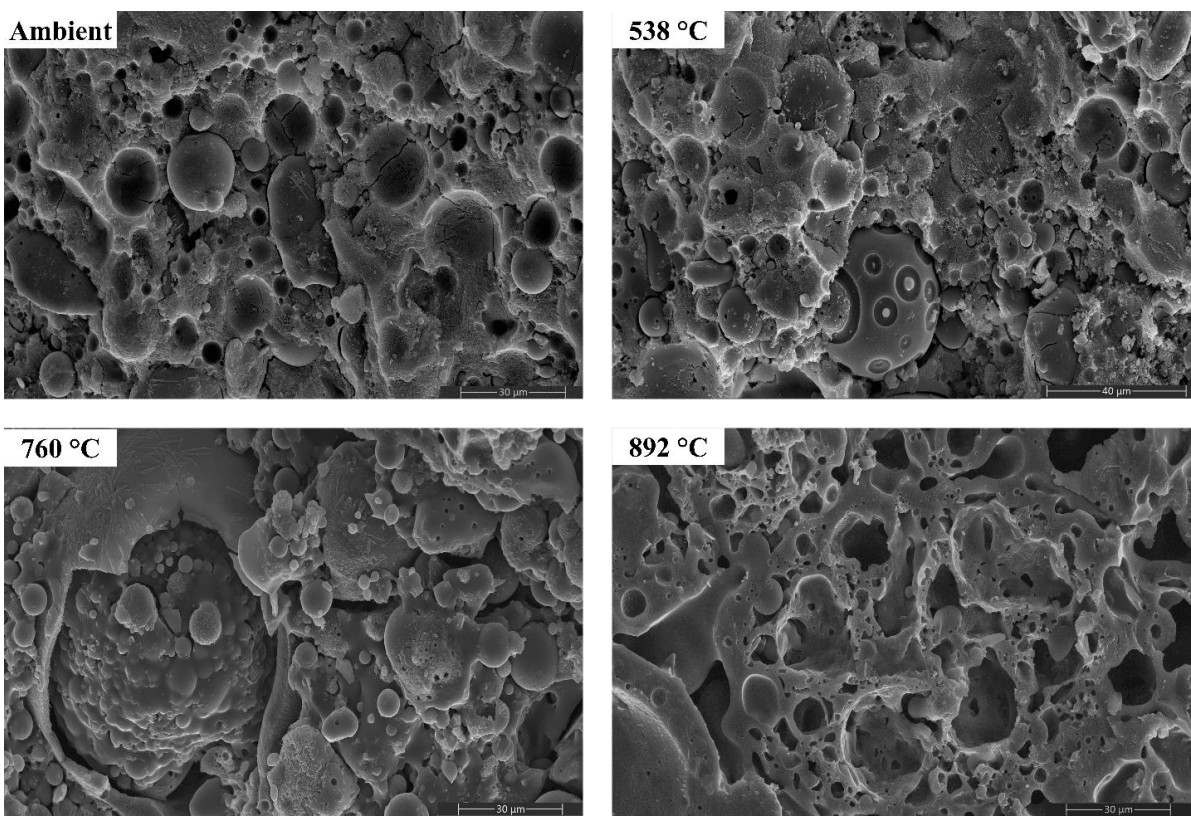

**Figure 15.** SEM micrographs of FS 0P-1.4 exposed to high temperatures.

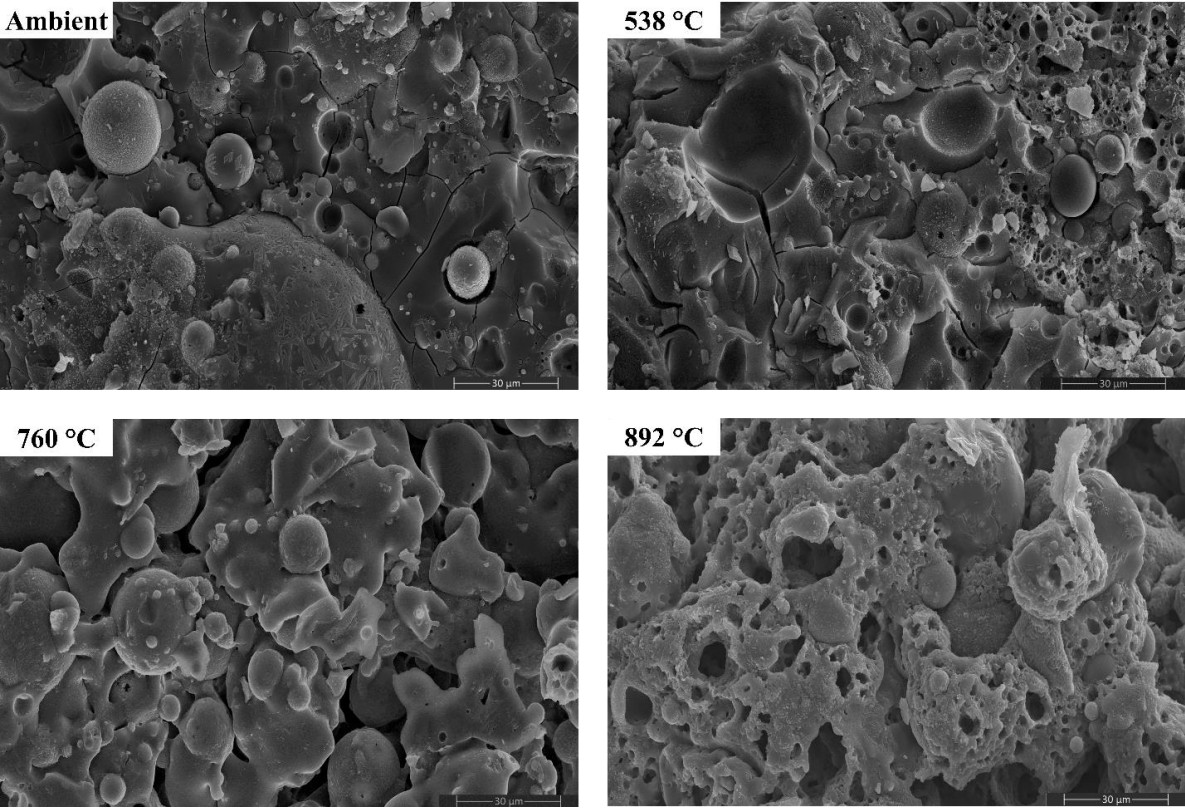

**Figure 16.** SEM micrographs of FS 30P-1.4 exposed to high temperatures.

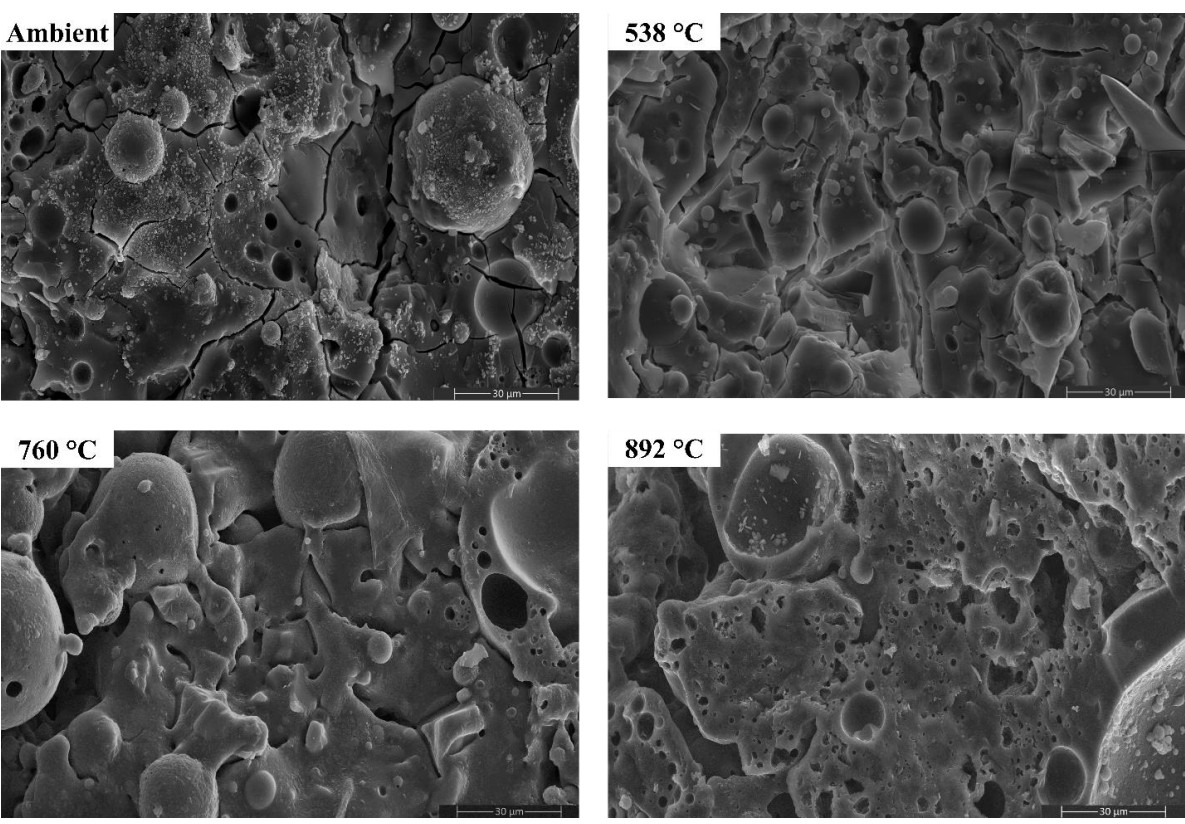

**Figure 17.** SEM micrographs of FS 40P-1.4 exposed to high temperatures.

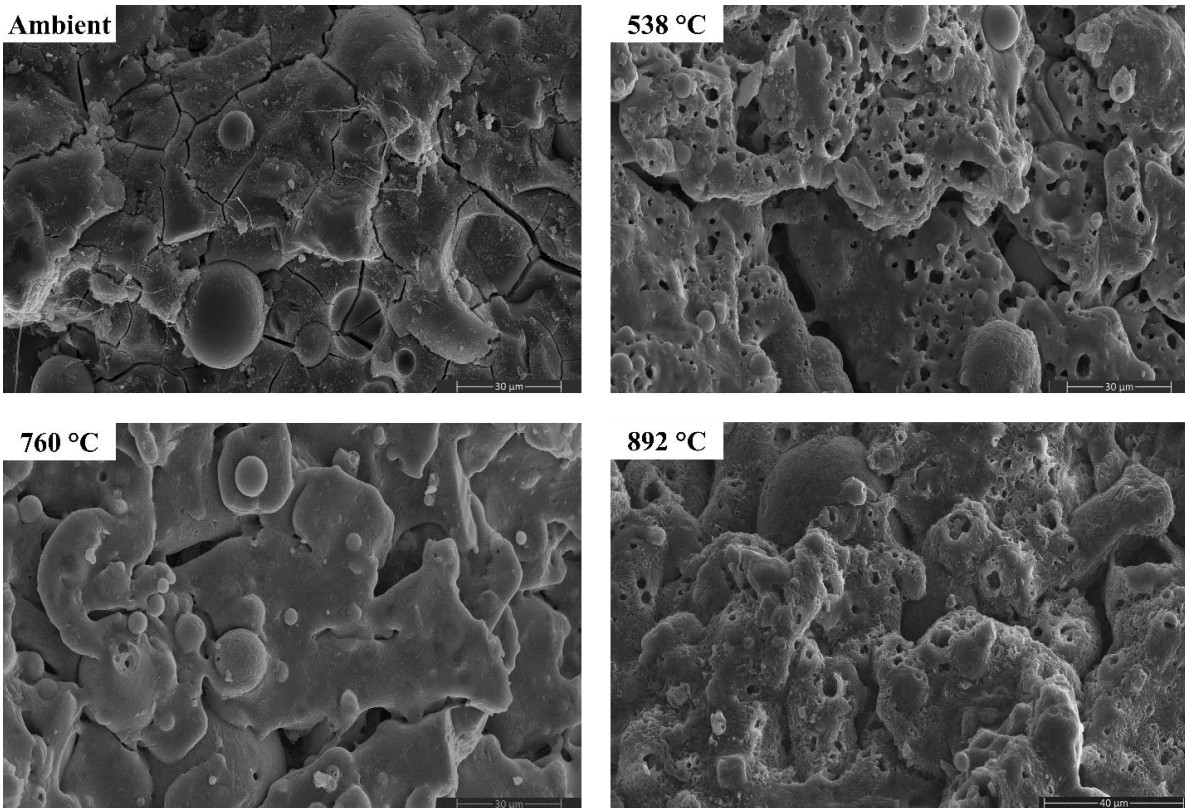

**Figure 18.** SEM micrographs of FS 50P-1.4 exposed to high temperatures.

One of the discernible features in the micrographs of AAB at 760 °C and 892 °C is the fusion and melting of the reaction products. FS 50P-1.4 evinced a denser matrix that corroborates the assumption that the denser matrix aggravates the pore pressure and deteriorates the performance. FS 0P-1.4 exhibits major changes at high temperatures. Pitting and collapse of fly ash particles are evident from the micrographs of FS 0P-1.4 at 892 °C, and the matrix resembled a near-molten mass. The vitreous phase observed from the micrographs of AAB paste at 760 °C and 892 °C consisted of small prismatic crystals and can be associated with the formation of akermanite, the presence of which is confirmed by XRD, which starts crystallization at temperatures beyond 760 °C.

As mentioned earlier, the results from SEM are supplemented with EDS results to quantify the chemical composition of the AAB microstructure. EDS analysis was performed at 45 different locations for each sample corresponding to four different mixes, i.e., FS 0P-1.4, FS 30P-1.4, FS 40P-1.4, and FS 50P-1.4. To distinguish the major reaction product formed in AAB with varying precursor proportions, the primary elements of the reaction products, viz. calcium, aluminum, and silicon were renormalized to 100% on an oxide basis and ternary phase diagrams are plotted (Figure 19).

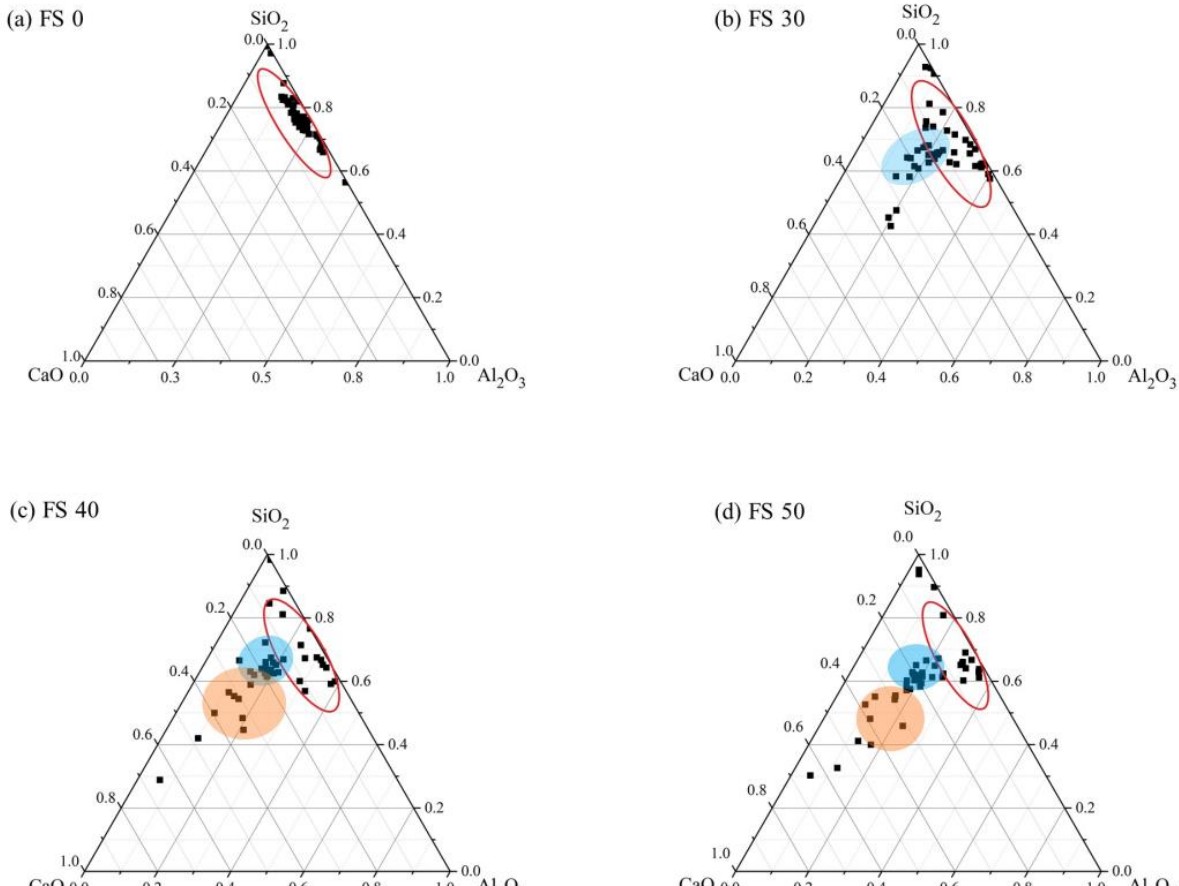

**Figure 19.** Ternary representation of EDS data for AAB with Ms of 1.4 exposed to ambient temperature.

These ternary diagrams show the apparent variation in the matrix composition besides the indication of their approximate chemical composition. The red ellipse indicates the N-A-S-H phase, the orange circle corresponds to the C-A-S-H phase, and the blue circle corresponds to the coexistence of both phases. The boundaries for these circles were established based on the existing literature [74,75]. Under ambient exposure conditions, alkali-activation of fly ash results in the formation of the N-A-S-H matrix (Figure 19a). From Figure 19b, it is observed that FS 30P-1.4 shows the formation of two distinct reaction prod-

ucts as a result of activation of both fly ash and slag, N-A-S-H and C-(A)-S-H. FS 40P-1.4 and FS 50P-1.4 reveal the coexistence of both the matrices (C-(N)-A-S-H). Based on the observations from previous studies, the compositional ranges of $0.72 < CaO/SiO_2 < 1.94$, $0 < Al_2O_3/SiO_2 < 0.1$ were used to identify the formation of C-A-S-H, whereas the compositional ranges of $0 < Na_2O/Al_2O_3 < 1.85$, $0 < CaO/SiO_2 < 0.3$, $0.05 < Al_2O_3/SiO_2 < 0.43$ were employed in the present study to demarcate C-(N)-A-S-H [65,74]. Existing research on FS 50P-1.4 mixes shows that their microstructural characteristics are similar to AAB in spite of the binding phase showing the coexistence of matrices. Ismail et al. (2014) [74] conclude that the C-(N)-A-S-H matrix in FS 50P-1.4 possess equivalent structural characteristics as C-A-S-H, and it is not possible to measure the chemical distinction between the two matrices using EDS [74]. Figures 20–23 present a ternary representation of EDS data of AAB with varying fly ash/slag ratios after exposure to high temperatures.

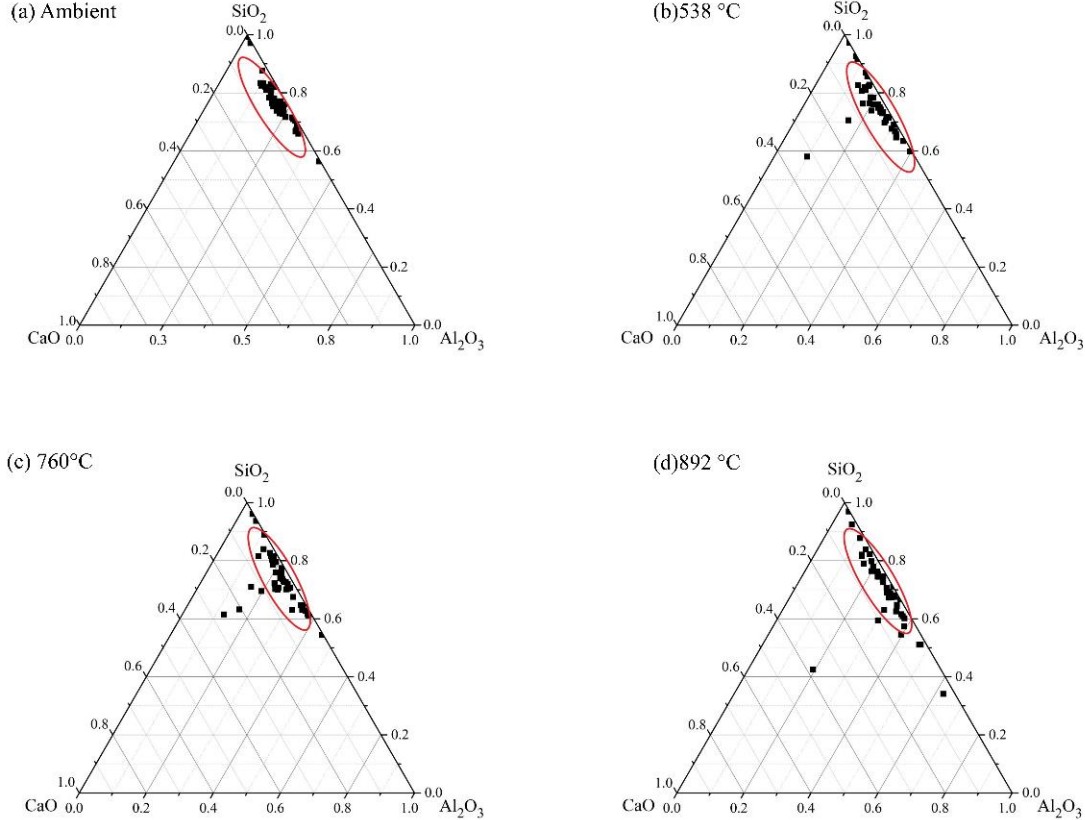

**Figure 20.** Ternary representation of EDS data for FS 0P-1.4 exposed to high temperatures.

An increase in the Al/Si ratio is associated with the conversion of amorphous reaction products to crystalline phases [75]. These findings are consistent with the XRD results, which show the formation of nepheline on exposure to elevated temperature (Figure 5). In FS 30P-1.4, FS 40P-1.4, and FS 50P-1.4, the Ca/Si ratio decreased with increasing temperature. The formation of crystalline phases of akermanite, and gehlenite, when exposed to temperatures above 750 °C, indicate the decomposition of amorphous reaction products. It is evident from the ternary phase diagram that the Ca/Si ratio increases with slag content; however, it decreases with the increase in temperature. The greater Ca/Si ratios of FS 40P-1.4 and FS 50P-1.4 at 892 °C compared to other mixes indicate the formation of greater amounts of akermanite and gehlenite, having Ca/Si ratios of 1 and 2, respectively.

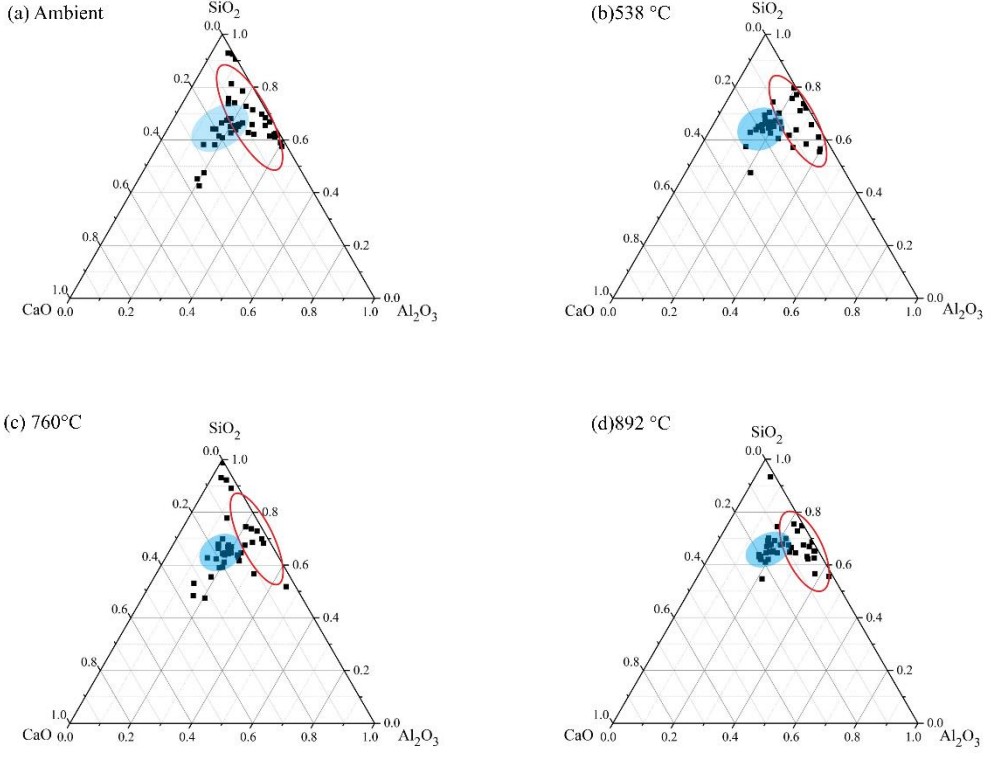

**Figure 21.** Ternary representation of EDS data for FS 30P-1.4 exposed to high temperatures.

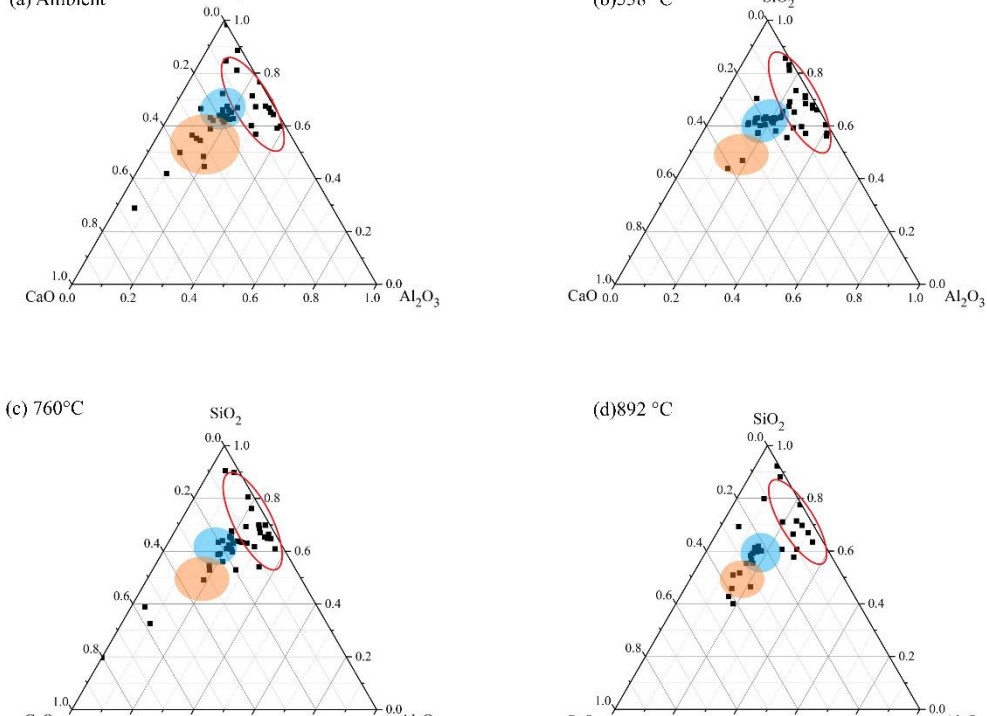

**Figure 22.** Ternary representation of EDS data for FS 40P-1.4 exposed to high temperatures.

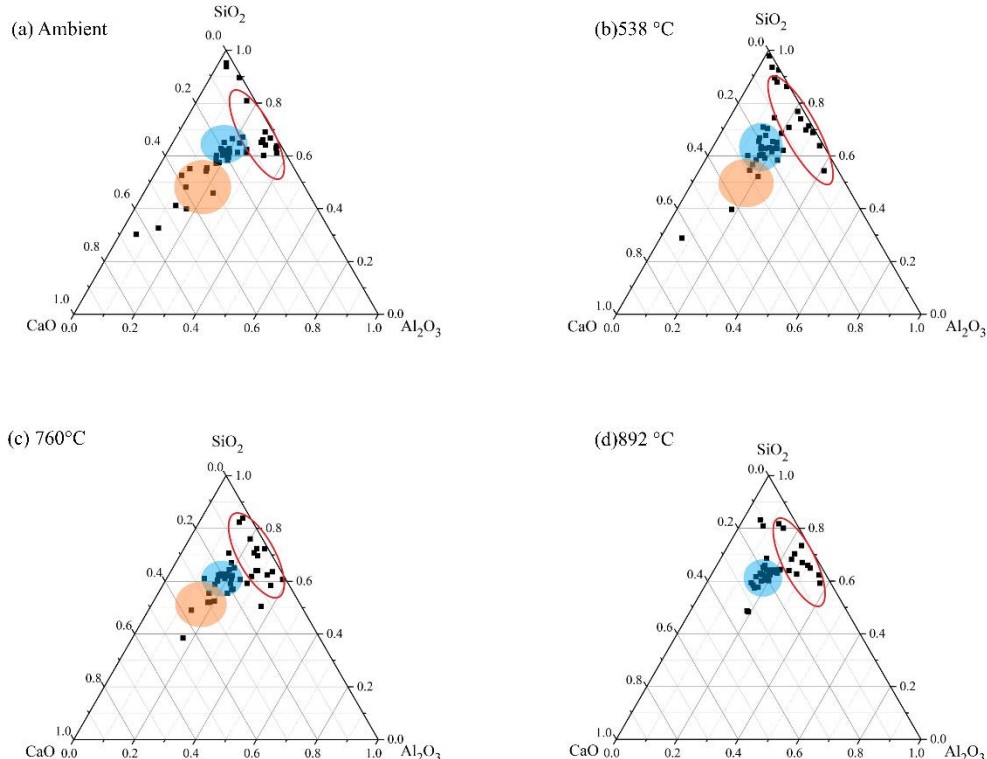

**Figure 23.** Ternary representation of EDS data for FS 50P-1.4 exposed to high temperatures.

### 3.4. Compressive Strength

The conspicuous observation from Figure 24 is the increase in strength with slag content, and this trend was exhibited by specimens having different Ms values. The increase in strength with increasing slag content is attributed to the formation of the additional reaction product, C-A-S-H. The increase in compactness of microstructure with slag content is associated with the formation of C-A-S-H. EDS analysis of the samples shows an increase in the formation of the C-A-S-H matrix with slag content. The significant effect of Ms on the compressive strength of AAC exposed to ambient temperature is evident from Figure 24. Lower Ms improved the strength of AAC and resulted in high early strength. This behavior is attributed to higher alkalinity, which increases the dissolution of the precursor particles, consequently increasing the rate of reaction [76].

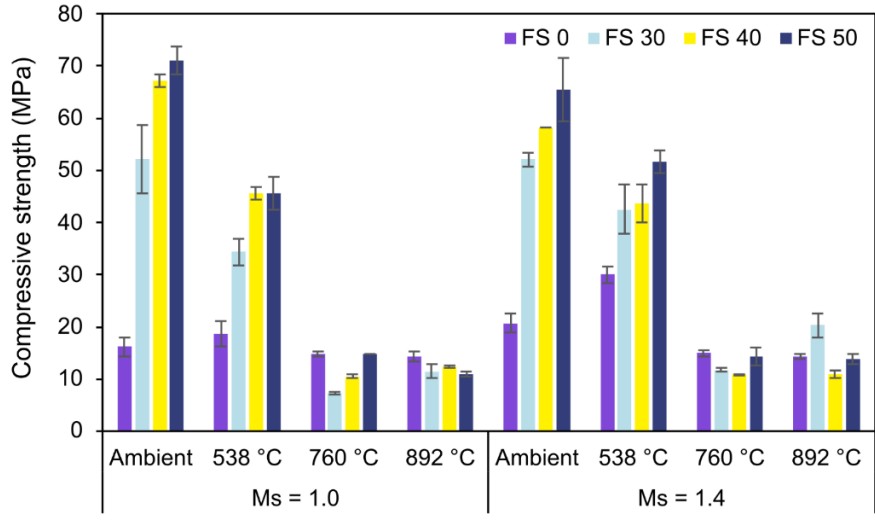

**Figure 24.** Compressive strength results.

With an increase in temperature to around 500 °C, further geopolymerization of the unreacted fly ash particles enhanced the strength. The microstructure plays a significant role in governing the mechanical performance of AAC, especially when exposed to high temperature. In cementitious systems exposed to high temperatures, it is observed that the pore pressure of water developed inside the specimens requires an escape route and otherwise develops stresses on the adjacent concrete resulting in spalling, explosion, or premature failure depending on its intensity [73]. This pore pressure of water is governed by the pore structure or the compactness of the microstructure. FS 50, which gives the highest strength at ambient exposure conditions, exhibits substantial loss in strength on exposure to 760 °C and 892 °C. This behavior is ascertained to its dense microstructure occluding the moisture escape from the specimen. This dense microstructure of FS 50 can be substantiated by the formation of C-A-S-H.

The compressive strength of FS 30 and FS 40 also decreased with an increase in temperature up to 760 °C. However, on exposure to a temperature of 892 °C, the compressive strength of both FS 30C-1.4 and FS 40C-1.4 increases. This behavior was attributed to the growth of crystalline phases of akermanite ($2CaO \cdot MgO \cdot 2SiO_2$) and gehlenite ($2CaO \cdot Al_2O_3 \cdot SiO_2$) in AAB when exposed to temperatures higher than 760 °C. According to available literature, slag was composed of a mixture of poorly crystalline phases with compositions resembling gehlenite and akermanite, with depolymerised calcium silicate glasses [77]. The growth of these crystals is promoted by high temperatures. These crystalline phases result in the formation of a stronger bond and result in the transition of the binder phase. The newly formed binder at this temperature is expected to re-establish the contact with aggregates improving the mechanical performance [78]. However, the reduction in strength validated the fact that the degradation due to pore pressure development dominates the enhancement of strength due to the formation of crystalline phases.

### 3.5. Pull-Out Test

The bond strength of AAC with Ms 1.0 and 1.4 and its behavior on exposure to high temperature is presented in Figure 25. The results presented here are the mean of three specimens assuming a uniform development of bond stress between the concrete and embedded rebar. Bond stress was calculated as the ratio of obtained load at failure and the surface area along the length of the embedded rebar.

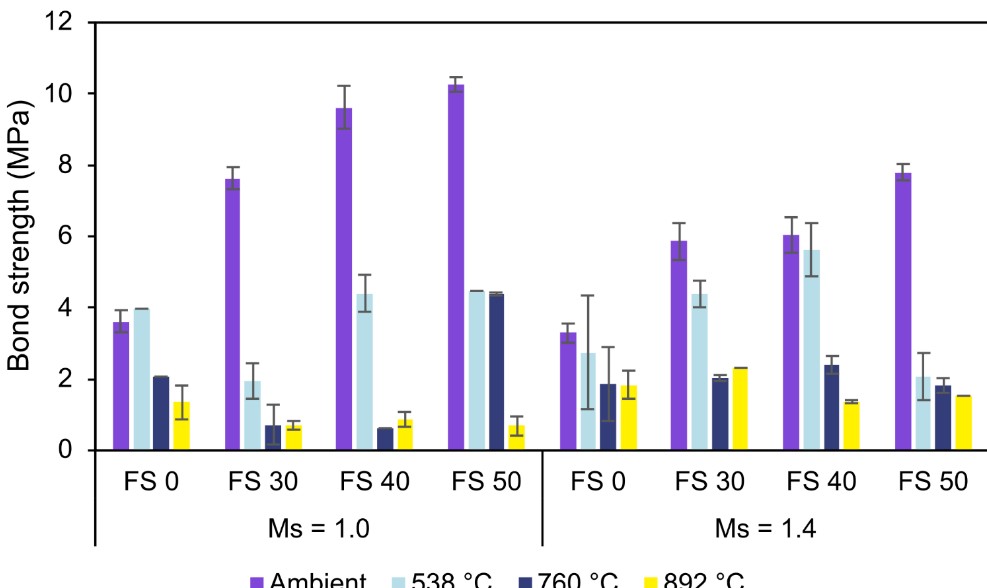

**Figure 25.** Bond behavior of AAC.

Analogous to the compressive strength results, the bond strength of AAC increased with slag content under ambient exposure conditions. The bond strength of FS 50 is 184.9% and 137.1% higher than FS 0 for Ms 1.0 and 1.4, respectively [79]. This high bond strength of FS 50 is attributed to the pore refinement resulting in a relatively uniform and compact ITZ between the embedded rebar and concrete. The improvement in strength and interfacial binding with slag content is attributed to a relatively higher proportion of dissolved species contributing to the formation of more reaction products [80]. The pore refinement in FS 50 was validated by the formation of an additional C-A-S-H matrix and a compact microstructure. It is well-known that the bond strength of cementitious systems depends on ITZ characteristics.

Unlike PC concrete with highly porous ITZ dominated by crystalline phases, ITZ in AAC is a dense and uniform gel-rich phase imparting strong interfacial bonding characteristics [81]. In AAB systems, the development of ITZ and its properties significantly depend on the reaction products and microstructural characteristics. The bond strength variation of AAC with Ms is equivalent to the trend observed in compressive strength test results. The higher bond strength of AAC mixes with Ms 1.0 is attributed to higher dissolved Al and Si species increasing the proportion of reaction products.

The bond strength of AAC decreased with increasing temperature. This depletion in the bond strength is higher in AAC with more slag content as the bond between the shorter C-A-S-H bonds deteriorates more than the bond between the rebars and the longer chain N-A-S-H bonds. However, if individually considered, the microstructure of the FS 30 is denser than the FS 0. Consequently, FS 30 exhibits the highest residual bond strength at 892 °C compared to other mixes. This behavior is attributed to the combined effect of the growth of crystalline phases (akermanite and gehlenite) densifying the matrix and the sintered fly ash particles increasing the pores and allowing evaporation of moisture.

*3.6. Flexural Strength*

The flexural strength of AAC with varying fly ash:slag ratio and Ms exposed to ambient temperature is presented in Figure 26. Flexural strength is a trivial mechanical property in analyzing the resistance of a beam or slab to bending failure. It is observed from the results that the flexural strength of AAC exhibits a trend equivalent to the compressive strength.

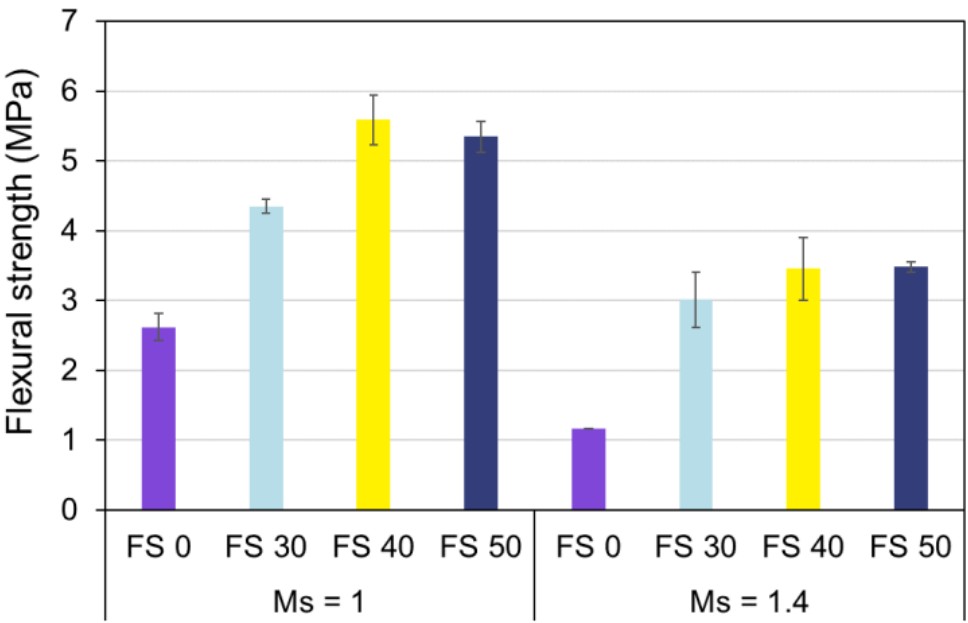

**Figure 26.** Flexural strength of AAC.

The flexural strength of AAC is dependent upon precursor proportions, and Ms. The flexural strength increased with slag content, and this is more pronounced in AAC with Ms of 1.0. Owing to greater alkalinity and subsequent rapid dissolution of precursors, the flexural strength of AAC with Ms of 1.0 is high.

*3.7. Split Tensile Strength*

The split tensile strength of AAC with varying Ms and precursor proportion under ambient exposure conditions is presented in Figure 27.

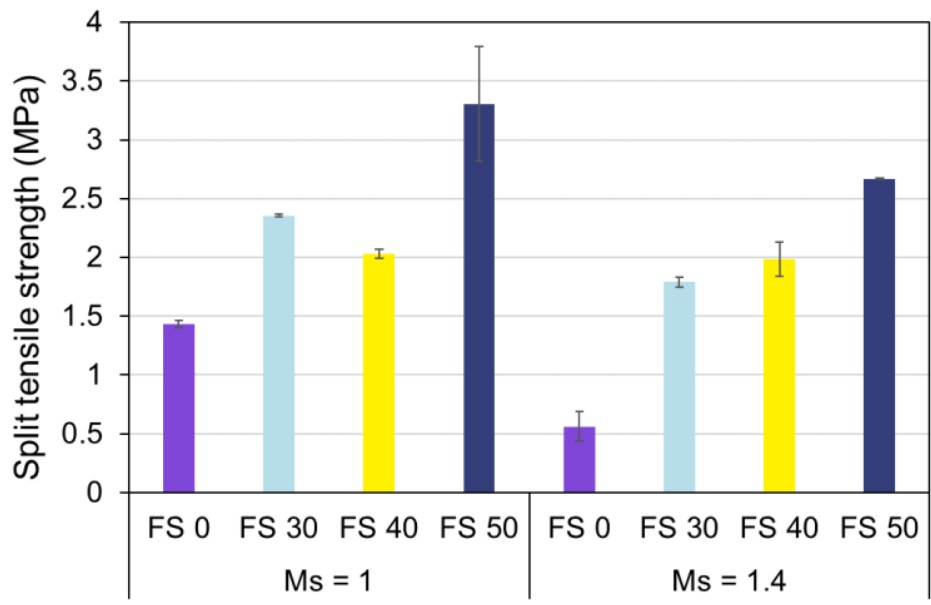

**Figure 27.** Split tensile strength of AAC.

The split tensile strength of AAC exhibited analogous behavior to compressive strength. The split tensile strength of AAC increased with slag content and decreased with Ms. It is evident that lower Ms enhances the split tensile strength of fly ash-based AAC owing to the enhanced dissolution of $Si^{+4}$ and $Al^{+3}$ species. Furthermore, the addition of 30% of slag (FS 30C-1.4) increased the split tensile strength by 219%.

*3.8. Life Cycle Assessment and Cost Analysis*

3.8.1. Midpoint Analysis

Figure 28 presents normalized impact categories of all mixes. It is conspicuous that PC concrete has approximately 1.5 times higher GWP than AAC, which is primarily related to the manufacturing process of PC. The impact categories related to ecotoxicity, eutrophication and ozone depletion have the highest contributions from AAC rather than PC concrete. This is attributed to alkaline activators used in AAC. Manufacturing processes of alkaline activators release toxic effluents, which are generally discharged into water or land bodies in the vicinity.

Furthermore, increasing Ms resulted in lower environmental impacts in all the damage categories. This implies that sodium hydroxide is a governing factor in affecting the environmental impacts. The chlor-alkali process of sodium hydroxide production emits $SO_2$ and halogenated compounds from electricity consumption during the electrolytic phase [82]. In the present study, chlor-alkali manufacturing through the diaphragm route was considered since it is the widely used method [83]. The chlor-alkali process of sodium hydroxide production has three different techniques viz., diaphragm cell, mercury cell, and membrane cell techniques. Of these, membrane cell is more sustainable owing to lower energy consumption and release of less toxic effluents compared to diaphragm and mercury cell process. In the diaphragm cell process, the primary toxic released is asbestos,

whereas, in the mercury cell process, it is mercury. Currently, the top producers of sodium hydroxide are western Europe, Japan, and the US. However, the majority of established plants established in western Europe are mercury cell, and in the US are diaphragm cell. In 2015, the European Union banned the use of mercury in the chlor-alkali industry [83].

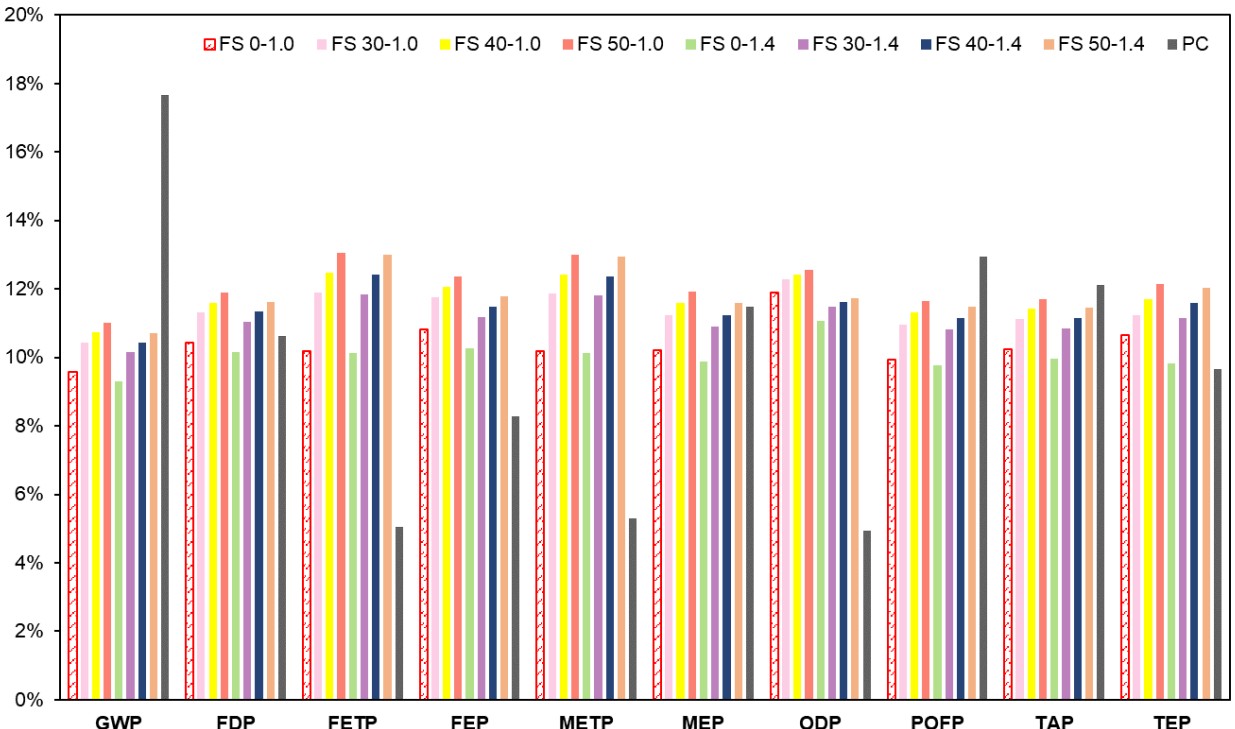

**Figure 28.** Normalized impact of AAC and PC concrete on midpoint damage categories.

The electrolytic phase of the chlor-alkali process using diaphragm technology is reported to consume 2.97 kWh/kg of electricity [84]. However, with increasing sustainability in the chlor-alkali industry, this calculation can be considered conservative for future production.

It is evident that AAC has a considerably lower impact on climate change. However, an improvement in the production process of alkaline activators will further enhance its environmental sustainability.

### 3.8.2. Endpoint Assessment

ReCiPe endpoint assessment has three areas of protection which are quality of ecosystem, human health, and resource scarcity. Figure 29 presents the endpoint assessment of AAC with varying precursor combination and Ms in comparison with PC concrete.

AAC mixes showed a lower or comparable effect on ecosystem quality. Increasing Ms exhibited a positive impact on the quality of the ecosystem and human health. PC concrete with an endpoint score of 20.7 has the highest impact on human health. Transportation contributes the most to resource depletion rather than the raw materials used to prepare AAC. Transportation resulted in 43–54% of the total resource depletion in AAC and 54% in PC concrete.

It is also observed that resource depletion is also contributed by slag in blended AAC mixes, and its contribution varied in the range of 17–25% of the total resource depletion, which is less than that caused by PC (37% with an endpoint score of 5.4 out of 14.8) in PC concrete. The resource depletion by slag can be attributed to both electricity and water consumption. It is emphasized here that in addition to the contribution by slag, the transportation distance of slag is the highest (431 km) in the present study. This aggravates the resource depletion by transportation. The PC production process has the highest effect on human health. The substances contributing to the detrimental effect on human health

are particulate matter (<2.5 μm) and N$_2$O, and the source of these substances is electricity consumption and clinker production [85].

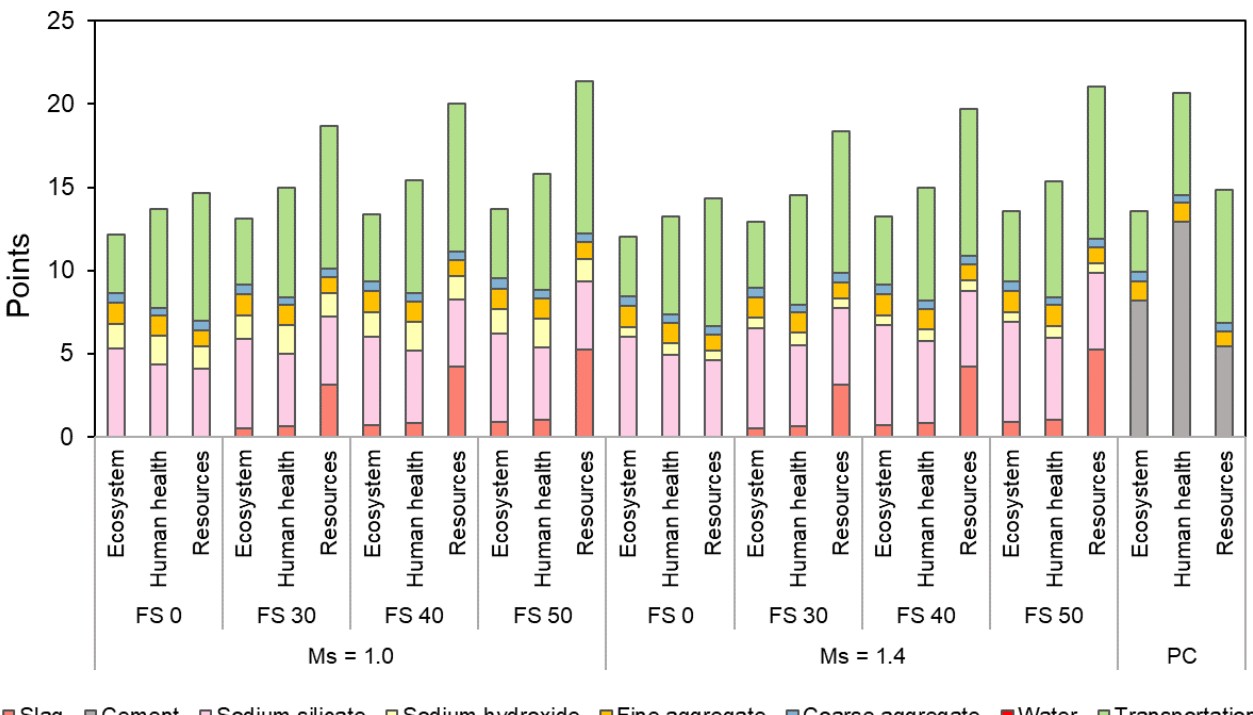

**Figure 29.** Endpoint environmental impact assessment of AAC and PC concrete.

International Reference Life Cycle Data System (ILCD) recommends default methods for different impact categories [46]. A similar reference model is used in ReCiPe for calculating the characterization factor (CF) as in ILCD default methods for climate change and ozone depletion. Consequently, as identified from Figure 30, the difference in the impact categories calculated using ReCiPe and IPCC 2013 for climate change and EDIP 2003 for ozone depletion is negligible. Though the underlying models are different in ReCiPe and CML 2001, the primary acidifying emissions considered are similar in the case of terrestrial acidification. This explains the minor variation observed in the calculated TAP using ReCiPe and CML 2001 methods.

However, the evident disparity in the toxicity potentials calculated using ReCiPe and USEtox for both freshwater ecotoxicity and human toxicity is attributed to the difference in their underlying models. The reason for the disparity between ReCiPe and USEtox is the variation in their respective inventory data. Additionally, since ReCiPe and USEtox do not share a common list of chemicals, a chemical may be linked to a CF in one database but not in the other. CFs are also unreliable, with a common operating presumption that they will differ by three orders of magnitude in any direction. For example, furan, which is a toxic organic compound (can be carcinogenic) and produced by the combustion of fossil fuels, has different CFs in ReCiPe and USEtox. In ReCiPe, the CF of furan is 2.14 1.4-DCB-Eq, whereas in USEtox, the CF for the same compound is 95.6 1.4-DCB-Eq. emitted to urban air affecting human health [45,48]. In comparison, the same compound has ReCiPe CF of $1.30 \times 10^{-6}$ 1.4-DCB-Eq. emitted to freshwater and USEtox CF of $4.08 \times 10^{-2}$ 1.4-DCB-Eq. emitted to freshwater. This elucidates the disparity of the impact categories calculated using these two methods. However, since USEtox is recommended as the default method by ILCD with a classification of II/III (II: recommended, some improvements needed; III: recommended, but to be applied with caution), it is suggested to use USEtox for the calculation of freshwater ecotoxicity and human toxicity potentials.

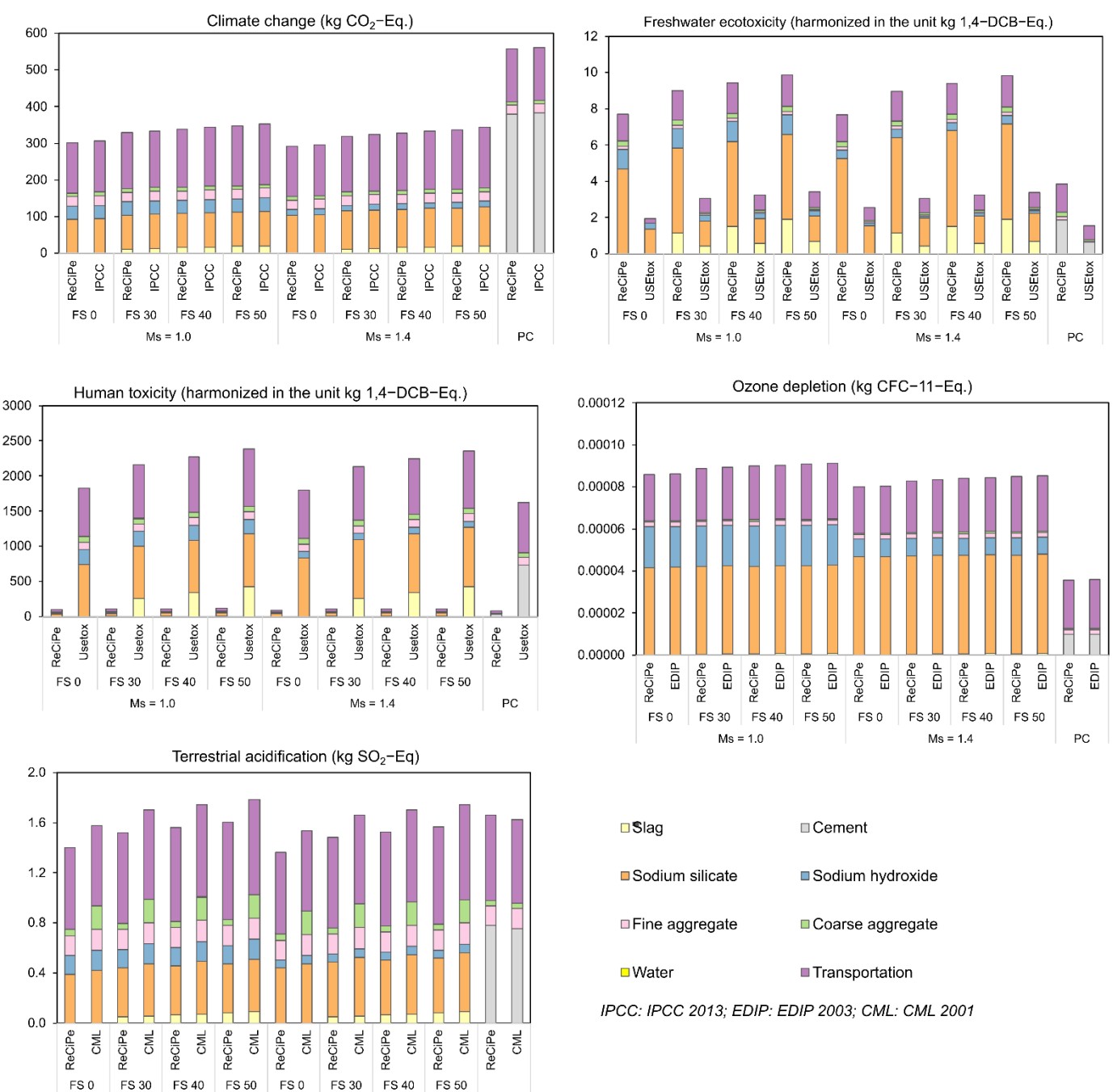

**Figure 30.** Comparison of environmental impacts using ILCD default methods.

### 3.8.3. Cost Analysis

Figure 31 presents the comparison of the cost of AAC with different precursor combinations, and Ms. It is evident that Ms 1.4 has an overall lower cost compared to Ms 1.0. The difference between production prices of AAC Ms 1.0 and Ms 1.4 varied in the range of 2.16–2.21%. Though it is not a huge difference when 1 m$^3$ is considered, however, this difference will regulate mix preference in case of mass concreting or large-scale constructions. The higher cost of AAC with Ms 1.0 is attributed to a lower content of sodium hydroxide in mixes with Ms 1.4. The unit cost (cost/kg) of sodium hydroxide is higher than sodium silicate.

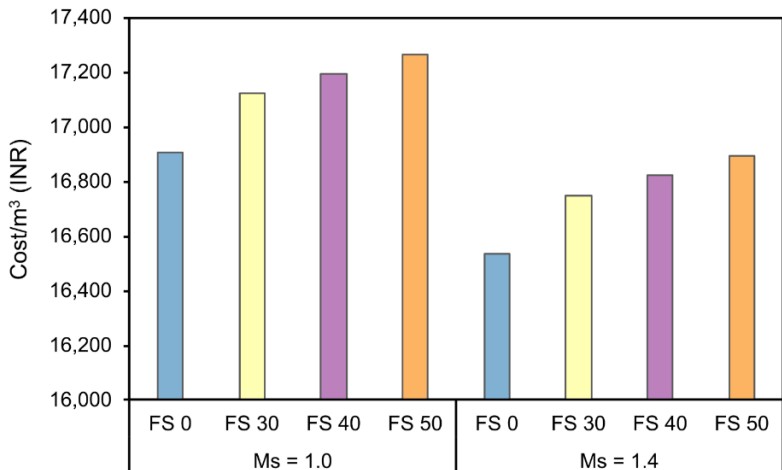

**Figure 31.** Cost estimate of AAC in the Indian context.

Based on the results from cost analysis, further studies on finding a suitable way to reduce the costs of sodium silicate and sodium hydroxide, either by adopting other alternative activators or developing an environmentally and economically viable production process, are recommended. Furthermore, a perspective on accounting for a carbon tax when cement is used is presented in the following section.

### 3.8.4. Effect of the Carbon Tax on the Cost of Concrete

The tax levied on GHG emissions associated with goods and services is known as the carbon tax. It is directly proportional to the sum of $CO_2$ emissions, typically expressed as a price per ton $CO_2$ equivalent (per ton $CO_2$-Eq.). Previous research on carbon tax policies, through a carbon tax, reports that over 40 countries and 16 provinces are currently collecting 'carbon revenues' [86].

To compare the effect of carbon pricing on the final cost of AAC and PC concrete, a critical situation of considering the highest carbon pricing (Sweden's carbon tax) is implemented in this study. Figure 32 presents a comparison of the effect of the carbon tax on the final cost of AAC and PC concrete production.

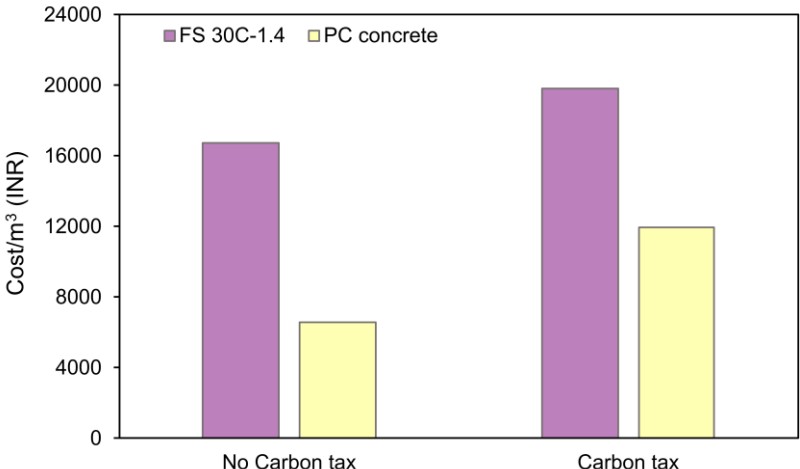

**Figure 32.** Effect of the carbon tax on the price of concrete.

From Figure 32, it is observed that when the carbon tax is implemented, the price of AAC increases by 18.40%, whereas the price of PC concrete increases by 81.68% [9]. This observation emphasizes the importance of reducing GHG emissions and the necessity of an environmentally and economically sustainable alternative binder.

## 4. Conclusions

In the present study, alkali-activated concrete with fly ash and slag as precursors was examined for its high-temperature performance and environmental impact. The specimen-level properties were correlated with the observed microstructural changes to have a better understanding of the underlying mechanism. The findings from the present study are summarized as follows:

- Exposure to the high temperature results in the formation of nepheline and albite in FS 0P-1.0 (fly ash:slag ratio of 100:0 and Ms of 1.0), and only nepheline in FS 0P-1.4 is confirmed through their corresponding XRD peaks, whereas blended alkali-activated binder mixes evinced formation of gehlenite, akermanite, and nepheline.
- The bands corresponding to N-A-S-H observed at 1013 cm$^{-1}$ in FS 0P shifted towards lower wavelengths evincing the formation of a combination of N-A-S-H and C-A-S-H matrices in blended alkali-activated binder. Consistent with XRD findings, formation of albite and nepheline in FS 0P was observed by the shift of peaks corresponding to the N-A-S-H matrix to higher wavelengths. The formation of nepheline and gehlenite in blended alkali-activated binder mixes were suggested by the shift of peaks at 991 to 1024 cm$^{-1}$ and 731 to 712 cm$^{-1}$, respectively.
- A higher proportion of unreacted fly ash particles owing to their high activation energy was observed in FS 0, and precipitation of different products with a smaller number of fly ash spheres was observed in blended alkali-activated binder through SEM micrographs.
- The percentage increase in 28-day compressive strength with slag content varied in the range of 151.8–339.7%, contingent on the proportion of added slag. High alkalinity and consequently enhanced dissolution of precursor particles led to improved compressive strength in alkali-activated concrete with activator modulus of 1.0.
- The increase in compressive strength of FS 30C and FS 40C on exposure to a temperature above 760 °C is attributed to the formation of akermanite and gehlenite, which re-establishes the contact with aggregates. The significant decrease in compressive strength of FS 50 mixes ranging between 85.4 and 89.1% is attributed to pore pressure development resulting in faster propagation of cracks.
- The decrease in bond strength of alkali-activated concrete with activator modulus varied in the range of 8.8–24.1% for FS 0 and FS 50 mixes, respectively, whereas the increase in bond strength with slag content was varied in the range of 111.4–184.9% for alkali-activated concrete with activator modulus of 1.0 and 78.2–137.1% for activator modulus of 1.4. The higher bond strength in FS 50C is attributed to a compact interfacial transition zone.
- FS 30C-1.4 exhibits superior high-temperature performance both in terms of residual compressive and bond strength. The flexural and split tensile strength of AAC followed a similar trend of compressive strength on varying activator modulus and precursor combination.
- The global warming potential of Portland cement concrete is 1.5 times that of alkali-activated concrete. Portland cement concrete with an endpoint score of 20.7 has the highest impact on human health owing to particulate matter and $N_2O$ emissions.
- The increased environmental impact related to ecotoxicity, eutrophication and ozone depletion by alkali-activated concrete is attributed to the manufacturing processes of alkaline activators. Using ReCiPe endpoint assessment, it is observed that increasing activator modulus resulted in a lower environmental impact on ecosystem quality and resource depletion.
- Transportation resulted in 43–54% of the total resource depletion in alkali-activated concrete and 54% in Portland cement concrete.
- The significant variation in freshwater ecotoxicity and human toxicity potentials computed using ReCiPe midpoint and USEtox models are due to different underlying models, variations in the inventory data, and their characterization factors. Hence, USEtox is recommended for evaluating the toxicity potentials.

- The cost per m$^3$ of alkali-activated concrete varied in the range of 16,532–17,265 INR (200–210 EUR). If a carbon tax is levied on greenhouse gas emissions, the increase in the cost of FS 30C-1.4 is 18.4%, whereas the cost of Portland cement concrete increases by 81.7%. This observation emphasizes the importance of reducing greenhouse gas emissions and the necessity of an environmentally and economically sustainable alternative binder.

**Author Contributions:** Conceptualization, K.K.R. and A.K.; methodology, K.K.R.; writing—original draft preparation, K.K.R.; writing—review and editing, A.K. and P.K.D.M.; supervision, A.K.; funding acquisition, A.K. All authors have read and agreed to the published version of the manuscript.

**Funding:** This research received no external funding.

**Institutional Review Board Statement:** This study did not require any ethical approval.

**Informed Consent Statement:** Not applicable.

**Data Availability Statement:** The data used for this study are reported in the manuscript. Any other information relevant to this study can be shared by the authors upon request.

**Conflicts of Interest:** The authors declare no conflict of interest.

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
