# Peer review of "High-Temperature, Bond, and Environmental Impact Assessment of Alkali-Activated Concrete (AAC)"

_infrastructures, doi:10.3390/infrastructures7090119_

Round 1

Reviewer 1 Report

The author studied experimentally the effect of varying fly ash: slag ratios, activator modulus, and high temperatures on the mechanical properties of AAC and microstructure of AAB. The paper is well organized and can be accepted after minor revision. Some suggestions:

- The abstract can be improved. The current version looks like a general introduction about the research work, especially the first half of the paragraph.

- Introduction part must be improved. It should contain a critical review of literature, a presentation of the research gap, emphasis on the article's novelty and originality. Hence, the authors are recommended to improve this part. The current version is too short in comparison with the whole lengthy article.

- Line 133, the sample size should be given.

- Figure 25 can be improved, especially the error bar.

Author Response

The abstract can be improved. The current version looks like a general introduction about the research work, especially the first half of the paragraph.

The abstract has been modified as per the suggestion of the reviewer.

Introduction part must be improved. It should contain a critical review of literature, a presentation of the research gap, emphasis on the article's novelty and originality. Hence, the authors are recommended to improve this part. The current version is too short in comparison with the whole lengthy article.

The introduction has been modified as per the suggestion of the reviewer.

Line 133, the sample size should be given.

The sample size was added to the modified version of the manuscript (Line 180).

Figure 25 can be improved, especially the error bar.

The figure has been improved as suggested by the reviewer.

Reviewer 2 Report

The paper “High-Temperature, Bond, and Environmental Impact Assessment of Alkali-Activated Concrete (AAC)” demonstrate an interesting study. However, some specific comments to improve the quality of the paper are below:

1.       In the abstract (lines 14-15), there are limited studies on the bond strength of ambient cured (fly ash + slag)-based alkali-activated concrete (AAC) exposed to high temperatures. Please rewrite the sentence to make a reader understand it properly. One suggestion is- Studies on the bond strength of fly ash and slag-based alkali-activated concrete (AAC) cured at ambient temperatures are inadequate when subjected to elevated temperatures.

2.       Line 28-29, the first line of the introduction should be a nicer and noncontradictory sentence. You can write as-Alkali-activated concrete (AAC), which is produced through chemical reactions between alkalis and aluminosilicate-rich precursors and is considered a potential replacement for typical Portland cement (PC) concrete.

3.       The introduction of the paper is too short, it should contain more recent research on AAC. AAC and/or geopolymer are not only environment friendly but they are also superior to PC concrete in many ways. Moreover, it is not only being researched but also used in many real-life applications around the world. One informative paper could be found here- https://doi.org/10.1016/j.conbuildmat.2019.117886

4.       Table 1. Specifications of precursors. (Supplied by the manufacturer), we should not rely on manufacturer data for chemical composition purposes.

5.       Line 232, The major crystalline phases of quartz (SiO2) and mullite (2Al2O3·SiO2) in the unreacted fly ash were observed in AAB, the findings of these minerals require more references. You can find some in the paper mentioned.

6.       3.8.3. Cost Analysis, the cost depends on the country you are considering, labour cost also varies with countries, you better mention here- in context of ……………(a country).

7.       Line 649, In the present study, AAC with fly ash and slag as precursors was examined for its high-temperature performance and environmental impact. Have you examined the environmental impact?

8.       Please remove the symbols and abbreviated terms from the conclusion as much as possible and make them plain and simple sentences to increase readability for all.  

9.       Indeed, it’s a good paper, and should be accepted after a minor revision.

Author Response

In the abstract (lines 14-15), there are limited studies on the bond strength of ambient cured (fly ash + slag)-based alkali-activated concrete (AAC) exposed to high temperatures. Please rewrite the sentence to make a reader understand it properly. One suggestion is- Studies on the bond strength of fly ash and slag-based alkali-activated concrete (AAC) cured at ambient temperatures are inadequate when subjected to elevated temperatures.

The abstract has been modified as per the suggestions of Reviewer 1. The authors thank Reviewer 2 for their insightful comments.

Line 28-29, the first line of the introduction should be a nicer and noncontradictory sentence. You can write as-Alkali-activated concrete (AAC), which is produced through chemical reactions between alkalis and aluminosilicate-rich precursors and is considered a potential replacement for typical Portland cement (PC) concrete.

The first sentence of the abstract/Intro is modified to include a non-contradictory sentence as per the suggestion of the reviewer.

The introduction of the paper is too short, it should contain more recent research on AAC. AAC and/or geopolymer are not only environment friendly but they are also superior to PC concrete in many ways. Moreover, it is not only being researched but also used in many real-life applications around the world. One informative paper could be found here- https://doi.org/10.1016/j.conbuildmat.2019.117886

Additional and relevant literature is added to the introduction of the revised version of the manuscript.

Table 1. Specifications of precursors. (Supplied by the manufacturer), we should not rely on manufacturer data for chemical composition purposes.

The unreliable information from Table 1 has been removed.

Line 232, The major crystalline phases of quartz (SiO2) and mullite (2Al2O3·SiO2) in the unreacted fly ash were observed in AAB, the findings of these minerals require more references. You can find some in the paper mentioned.

The authors would like to highlight that XRD analysis for raw fly ash and slag has been performed. The crystalline phases of mullite and quartz present in fly ash were evaluated and detected by the current authors, phase identification was performed through HighScore Plus software as mentioned in the manuscript.

3.8.3. Cost Analysis, the cost depends on the country you are considering, labour cost also varies with countries, you better mention here- in context of ……………(a country).

The authors agree with and thank the reviewer for highlighting an important issue. The country based on which cost analysis was performed is added to Figure 31 title.

Line 649, In the present study, AAC with fly ash and slag as precursors was examined for its high-temperature performance and environmental impact. Have you examined the environmental impact?

The environmental impact in the present study is evaluated through life cycle assessment and the results for the same are presented in section 3.8.

Please remove the symbols and abbreviated terms from the conclusion as much as possible and make them plain and simple sentences to increase readability for all. 

As suggested by the reviewer the abbreviations were removed, however, the mix denominations were retained since they have been maintained uniformly throughout the manuscript.

Reviewer 3 Report

This study evaluates the effect of varying fly ash: slag ratios, activator modulus (Ms), and high temperatures on the mechanical properties of AAC and microstructure of AAB. The experiments are well designed and sufficient. The conclusions are well supported by the results. I believe this is generally a good research. Below are my suggestions.

1. The literature review is not enough. The recent studies on the thermodynamic modelling of alkali-activated materials should be included. Based on the thermodynamic simulation, the hydration products of AAMs can be predicted theoretically. ("Analytical investigation of phase assemblages of alkali-activated materials in CaO-SiO2-Al2O3 systems: The management of reaction products and designing of precursors. Materials & Design194, p.108975.").

2. The innovation of this study should be highlighted and the research gap should be explained in detail.

3. The conclusions are not very attractive. More general and scientific conclusions should be summarized about the influence of Ms, raw materials etc. on the properties of materials. The current conclusions are more like those in a project report for specific materials.

Author Response

The literature review is not enough. The recent studies on the thermodynamic modelling of alkali-activated materials should be included. Based on the thermodynamic simulation, the hydration products of AAMs can be predicted theoretically. ("Analytical investigation of phase assemblages of alkali-activated materials in CaO-SiO2-Al2O3 systems: The management of reaction products and designing of precursors. Materials & Design, 194, p.108975.").

Additional and relevant literature is added to the introduction of the revised version of the manuscript.

The innovation of this study should be highlighted and the research gap should be explained in detail.

The last paragraph of the introduction highlights and discusses, in brief, the research gap addressed and the novelty of the present study.

The conclusions are not very attractive. More general and scientific conclusions should be summarized about the influence of Ms, raw materials etc. on the properties of materials. The current conclusions are more like those in a project report for specific materials.

The current study includes only two different precursors and activator modulus. The parameters varied in the current study are small to make generic conclusion statements. The conclusions presented are specific to the current study and hence the observed results were summarized instead of generic statements.

Round 2

Reviewer 3 Report

It is claimed that "the results from microstructural experiments show the formation of new crystalline phases and decomposition of reaction products on high temperature exposure, and they correlate well with the observed mechanical performance."

In fact, the the high temperature exposure will not cause the formation of crystalline phases since it's more related to the cooling process. If you let it cool down slowly, the crystalline phases will definitely form.

How do they correlate well with each other? Have you done micromechanical modelling?

Round 3

Reviewer 3 Report

This paper has been revised based on the comments.